JCB Journal of Cell Biology

# Polar relaxation by dynein-mediated removal of cortical myosin II

Bernardo Chapa-y-Lazo[1] , Motonari Hamanaka[1,2], Alexander Wray[1,3], Mohan K. Balasubramanian[1] , and Masanori Mishima[1] 

**Nearly six decades ago, Lewis Wolpert proposed the relaxation of the polar cell cortex by the radial arrays of astral microtubules as a mechanism for cleavage furrow induction. While this mechanism has remained controversial, recent work has provided evidence for polar relaxation by astral microtubules, although its molecular mechanisms remain elusive. Here, using *C. elegans* embryos, we show that polar relaxation is achieved through dynein-mediated removal of myosin II from the polar cortexes. Mutants that position centrosomes closer to the polar cortex accelerated furrow induction, whereas suppression of dynein activity delayed furrowing. We show that dynein-mediated removal of myosin II from the polar cortexes triggers a bidirectional cortical flow toward the cell equator, which induces the assembly of the actomyosin contractile ring. These results provide a molecular mechanism for the aster-dependent polar relaxation, which works in parallel with equatorial stimulation to promote robust cytokinesis.**

## Introduction

During animal cell cytokinesis, cleavage by constriction of an actomyosin contractile ring is spatially coupled to chromosome segregation by the mitotic apparatus (Green et al., 2012; D'Avino et al., 2015). The best understood mechanism for spatial coupling is chemical signaling via activation of the RhoA GTPase at the cell equator by the central spindle, which is mediated by centralspindlin, a microtubule-bundling signaling hub, and ECT2 RhoGEF under the regulation of mitotic kinases (White and Glotzer, 2012; Mishima, 2016). A phosphatase-mediated negative signal from the kinetochores has also been reported to reduce the F-actin levels at the polar cortexes (Rodrigues et al., 2015). However, in cells with a relatively small mitotic spindle such as the *Caenorhabditis elegans* one-cell stage embryo, neither positive nor negative signals from the spindle or chromosomes alone can effectively work on the distant cell cortex at the initial stage of cytokinesis before the cleavage furrow deepens (von Dassow, 2009; Mishima, 2016). In these cells, signals from the astral microtubules, both positive ones to the cell equator and negative ones to the polar/nonequatorial cortexes, play key roles.

Recent studies suggest that the positive signaling to the equator ("equatorial stimulation") via astral microtubules uses a similar mechanism to the one from the central spindle, mediated by centralspindlin (D'Avino et al., 2006; Nishimura and Yonemura, 2006; Argiros et al., 2012; Nguyen et al., 2014; Su et al., 2014; Uehara et al., 2016). As another mechanism, the idea of furrow induction by relaxation of the polar cortexes via a

signal from the astral microtubules ("astral relaxation/polar relaxation") has a longer history (Wolpert, 1960; White and Borisy, 1983), although early experimental work failed to support this theory (Rappaport, 1996). However, more recently, evidence has been accumulating for the suppression of the contractility of the polar cortexes by radial arrays of astral microtubules (polar relaxation) in various cell types, including *C. elegans* embryos Dechant and Glotzer, 2003; Werner et al., 2007; Khaliullin et al., 2018), echinoderm embryos (Foe and von Dassow, 2008; von Dassow et al., 2009), silkworm spermatocytes (Chen et al., 2008), and mammalian cultured cells (Murthy and Wadsworth, 2008). Polar relaxation contributes to furrow formation often by promoting the flow of the actomyosin network within the cell cortex (cortical flow; Dan, 1954; Werner et al., 2007; Chen et al., 2008; Murthy and Wadsworth, 2008; Khaliullin et al., 2018), which contributes to the assembly of the actomyosin contractile ring (DeBiasio et al., 1996; Zhou and Wang, 2008; Salbreux et al., 2009; Turlier et al., 2014; Reymann et al., 2016) and releases cytoplasmic hydrostatic pressure, which would otherwise destabilize the furrow (Sedzinski et al., 2011). However, the molecular mechanisms responsible have remained unclear.

An important action of astral microtubules is to generate the mechanical forces that pull the spindle poles toward the cell cortex through the interaction with the cortically anchored subpopulations of dynein, a multifunctional minus-end directed motor (Grill et al., 2001; Pecreaux et al., 2006; Kozlowski et al.,

[1]Centre for Mechanochemical Cell Biology & Division of Biomedical Sciences, Warwick Medical School, Coventry, UK;   [2]Hokkaido University, Sapporo, Japan;   [3]University of Nottingham, Nottingham, UK.

Correspondence to Masanori Mishima: m.mishima@warwick.ac.uk;   Mohan K. Balasubramanian: m.k.balasubramanian@warwick.ac.uk.

2007; Saunders et al., 2007; Panbianco et al., 2008; Kotak et al., 2012; Kiyomitsu and Cheeseman, 2013; Lee et al., 2015; Maton et al., 2015; Nahaboo et al., 2015; Schmidt et al., 2017; Fielmich et al., 2018). After anaphase onset, more astral microtubules start to grow toward the cell cortex and, in one-cell-stage *C. elegans* embryos, interact first with the posterior cortex and then with the anterior cortex to generate cortical pulling forces. These dynein-mediated cortical pulling forces drive asymmetric spindle positioning and spindle elongation in coordination with microtubule-sliding and polymerization at the spindle midzone (Saunders et al., 2007; Lee et al., 2015; Maton et al., 2015; Nahaboo et al., 2015). Although defects in cortical force generation alone do not prevent cleavage furrow formation, combinations with central spindle defects severely do (Dechant and Glotzer, 2003; Bringmann et al., 2007; Verbrugghe and White, 2007; Werner et al., 2007). This indicates that a dynein-mediated mechanism contributes to cleavage furrow formation. However, it remains unclear whether this is due to an indirect effect of the altered spindle geometry or a more direct influence of dynein and astral microtubules on the activities of the cortical actomyosin network.

In this manuscript, in which the term polar is used to refer to the nonequatorial zone, we report internalization of myosin II from the polar cortexes by astral microtubules and dynein, and investigate the possibility that this activity triggers the polar relaxation and facilitates the contractile ring assembly via the induction of a bidirectional cortical flow toward the cell equator.

## Results

### Dynein drives unidirectional transport of myosin II along astral microtubules during anaphase

Nonmuscle myosin II is a major component of the cortical actomyosin network and the cytokinetic contractile ring, and is crucial for animal cell cytokinesis. Myosin dynamics in dividing *C. elegans* embryos has previously been studied, but data are limited to a low temporal resolution (greater than ~10 s) for a 3D volume (Carvalho et al., 2009; Jordan et al., 2016; Khaliullin et al., 2018) or, to higher temporal resolution, but only at the cell surface (Werner et al., 2007; Reymann et al., 2016). To examine the rapid dynamics of myosin II on the mitotic spindle and astral microtubules, we performed fast time-lapse recording (approximately every second) of GFP-tagged nonmuscle myosin II (NMY-2::GFP), expressed from the endogenous locus (*nmy-2(cp13)*; Dickinson et al., 2013), imaging a 2-μm-thick z-section at the embryo midplane with 0.5-μm z-steps. Embryos within the eggshell were immobilized on the surface of a coverglass without any deformation to avoid the influence of mechanical stress on the cortical actomyosin network (Singh et al., 2019). During metaphase, in addition to accumulation at the cell cortex, NMY-2::GFP was detected on the spindle (Fig. 1 A, metaphase). NMY-2::GFP was also observed as cytoplasmic particulate signals, which showed diffusive random motion and gradually disappeared during early mitosis and were nearly undetectable at the metaphase to anaphase transition (Fig. 1 A, metaphase; Videos 1 and 2). By contrast, after anaphase onset, unidirectional movement of cytoplasmic myosin II particles toward the spindle

poles was detected (Fig. 1 A, anaphase; Videos 2 and 3), with a mean velocity of ~0.7 μm/s (Fig. 1, B and C). Anillin, a multifunctional cytoskeletal protein that interacts with myosin II and other contractile ring components (Field and Alberts, 1995; Oegema et al., 2000; Maddox et al., 2007; Piekny and Glotzer, 2008; Piekny and Maddox, 2010; Tse et al., 2011), also showed a similar unidirectional movement toward the spindle poles (Fig. 1 D) although, in this case, such movement was less prominent than that of the particulate signals that are immobile relative to the spindle poles. Overlaying the trajectories of myosin II from multiple embryos revealed a bipolar spatial pattern similar to that of the spindle asters (Fig. 1 E).

The direction and velocity of the cytoplasmic movement of myosin II particles were consistent with motility by dynein, a microtubule minus-end-directed motor (Reck-Peterson et al., 2018). To test the role of microtubules in the myosin II particle movement, embryos were treated with a microtubule depolymerizing drug, nocodazole (Fig. 2 A). As expected, the cytoplasmic movement of myosin II particles was quickly suppressed after nocodazole treatment (5/5 embryos), while it was not affected at all in control embryos treated with the drug vehicle (4/4). To test the role of dynein in the directional transport of myosin II as its potential cargo, we used RNAi to deplete embryos of the dynein heavy chain DHC-1 (Gönczy et al., 1999) or to deplete LIN-5, the orthologue of NuMA, a dynein regulator (Lorson et al., 2000; Radulescu and Cleveland, 2010; Fig. 2 B). We also tested ciliobrevin D (Firestone et al., 2012), an inhibitor of vertebrate cytoplasmic dynein, but this did not cause any mitotic or developmental abnormalities (data not shown). No unidirectional cytoplasmic movement of myosin II was observed in the *dhc-1(RNAi)* embryos (7/7 embryos; Fig. 2 B, *dhc-1(RNAi)*; Video 4), which underwent massively disorganized cell division reflecting the multiple mitotic functions of dynein in centrosome separation, metaphase chromosome congression, spindle positioning and elongation, and chromosome segregation (Gönczy et al., 1999). In *lin-5(RNAi)* embryos, mitosis proceeded more normally except for expected defects in spindle positioning and elongation (Srinivasan et al., 2003; Panbianco et al., 2008; Galli et al., 2011; Lee et al., 2015). As in the case of *dhc-1(RNAi)* embryos, no cytoplasmic unidirectional movement of myosin II particles was detected in the embryos depleted of LIN-5 (33/33 embryos; Fig. 2 B, *lin-5(RNAi)*; Video 5) or its binding partners GPR-1/2, the Pins/LGN orthologues (Colombo et al., 2003; Gotta et al., 2003; Srinivasan et al., 2003; videos not shown, quantitative analysis in Fig. 7 C to be described in a later section). These results indicate that the movement of myosin II from the cortex toward the spindle poles depends on astral microtubules and is driven by dynein.

Interestingly, we observed a significant delay in the furrow formation in the embryos depleted of LIN-5 or GPR-1/2 and thus lacking the internalization of the myosin II particles (Fig. 2 C). While the first sign of furrow ingression was detected by visual inspection at ~150 s post-anaphase onset (p.a.o.) in control embryos, it was delayed by more than 40 s in the *lin-5(RNAi)* and *gpr-1/2(RNAi)* embryos (Fig. 2 C). These data are consistent with a positive role of dynein-mediated myosin II transport in cleavage furrow formation.

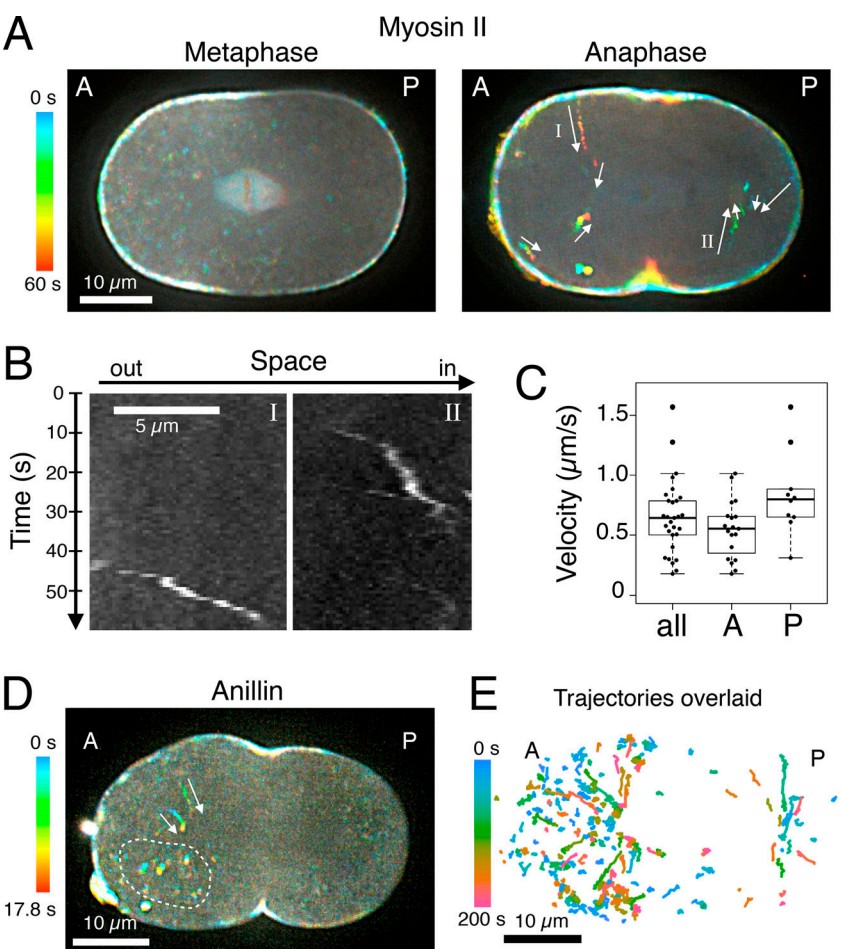

Figure 1. **Centrosome-directed movement of myosin II particles along astral microtubules during anaphase.** (A) GFP-tagged NMY-2 in the mid-plane of *C. elegans* embryos was imaged by spinning disk confocal microscopy every 0.83 s and is presented with temporal color coding. Unidirectional movement of the myosin II in the cytoplasm (arrows) was observed during anaphase. (B) Kymographs of the myosin II particles labeled I and II in the anaphase embryo in A. (C) Velocity distributions of the unidirectional movement of the cytoplasmic myosin II particles found in anterior (A) and posterior (P) sides of anaphase embryos (*n* = 19 and 10, respectively, from four embryos). (D) Anillin::GFP in the mid-plane was imaged every 0.72 s and is presented with temporal color coding. Unidirectional transport of anillin similar to that of myosin II was detected (arrows) although particles immobile relative to the spindle poles were observed more frequently (e.g., those surrounded by a dotted curve). (E) The trajectories of the cytoplasmic transport of myosin II particles from 22 embryos were overlaid and temporally color-coded based on the time after anaphase onset.

## Myosin II removal from the cortex by dynein and astral microtubules is associated with the local relaxation of the cortical actomyosin network

The aster-dependent unidirectional motion of myosin II particles toward the spindle poles suggests that they are derived from the myosin II population that has been accumulated at the cell cortex but then has been removed from it by dynein and the astral microtubules. It has been reported that dynein and the astral microtubules cause invagination of the plasma membrane during anaphase, which becomes exaggerated upon disruption of the cortical actomyosin network by depletion of myosin II or by treatment with cytochalasin D, an actin inhibitor (Redemann et al., 2010; Tse et al., 2011). To examine the relationship between the internalization of myosin II and the invagination of the plasma membrane, we simultaneously observed NMY-2::tagRFP-T (myosin II tagged with a red fluorescent protein; Nishikawa et al., 2017) and PH(PLCδ1)::GFP (a plasma membrane marker based on a pleckstrin homology (PH) domain; Audhya et al., 2005) in the midplane of an embryo in anaphase. Some membrane invaginations were led by a myosin II particle (Fig. 3 A; and Video 6, arrow), while no myosin II peak was detected at the leading tip of other invaginations (Fig. 3 A; and Video 6, arrowhead). Colocalization of myosin II and the PH domain was detected on some cytoplasmic particles (Fig. 3 B; and Video 7, white arrowhead) but not on other particles (Fig. 3 B; and Video 7, magenta and green arrowheads). We examined eight PH/

myosin dual-labeled embryos and observed 20 myosin II particles negative for the PH domain signal, 31 membrane signals positive for the PH domain but negative for myosin II, and eight double positive particles. According to these observations, it is likely that dynein and astral microtubules act on the plasma membrane and the myosin II particles through distinct mechanisms. Although it is possible that internalization of other factors might play a role in the regulation of cytokinesis, we focused on the removal of the myosin II particles from the cell cortex in this work.

To further examine the action of the astral microtubules on the cortical actomyosin network, we next observed embryos expressing both NMY-2::tagRFP-T (Nishikawa et al., 2017) and tubulin::YFP (Kozlowski et al., 2007; Fig. 3, C–F; and Video 8). We detected the internalization of single myosin II particles from the cell cortex at the site of contact with single astral microtubule fibers (Fig. 3, C and E; and Videos 9 and 10, posterior and anterior cortexes, respectively). A particle (red arrowheads) traveled along an astral fiber toward the associated spindle pole, although we usually failed to track the particle until it reached the pole due to the focal drift of the spindle pole or the disappearance of the particle. Following the internalization of a myosin II particle, gradual separation of flanking peak signals of the cortical myosin II was observed (Fig. 3, D and F; and Videos 9 and 10, blue arrowheads). Although it was difficult to systematically and statistically assess the frequency of this phenomenon, in 20

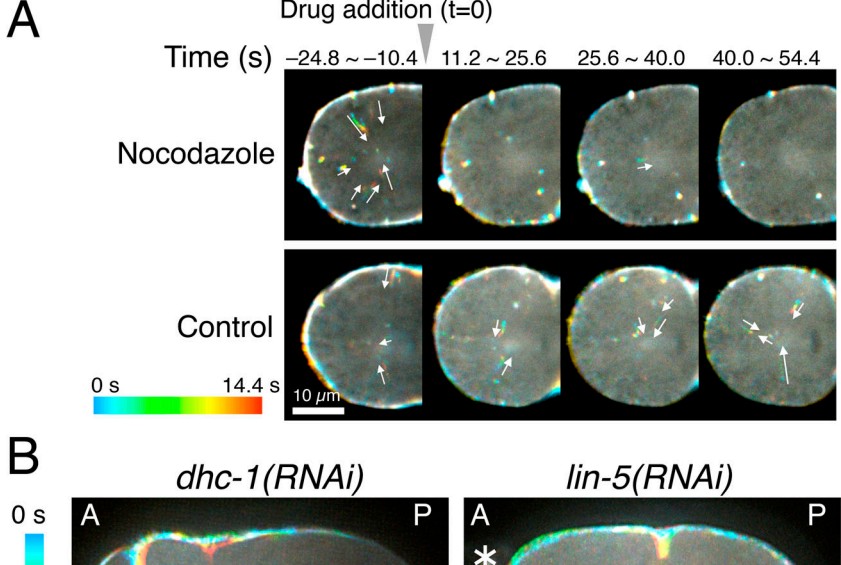

Figure 2. **Cytoplasmic transport of myosin II particles and timely furrow formation depend on the astral microtubules and dynein.** **(A)** Embryos expressing NMY-2::GFP were treated with 15 µM nocodazole, a microtubule poison, or the drug vehicle (control) during anaphase (at 0 s). Nocodazole strongly suppressed the cytoplasmic motility of myosin II. **(B)** Depletion of the cytoplasmic dynein heavy chain (*dhc-1(RNAi)*) or the dynein regulator LIN-5/NuMA (*lin-5(RNAi)*) eliminated the cytoplasmic unidirectional movement of myosin II. * indicates the polar body. **(C)** Timing of the initial sign of furrow formation determined by visual inspection of anonymized videos (mean ± SEM). Statistical significance was tested against the control by linear modeling with correction for multiple comparisons by Dunnett's method. *, **, and *** indicate P < 0.02, 0.01, and 0.001, respectively.

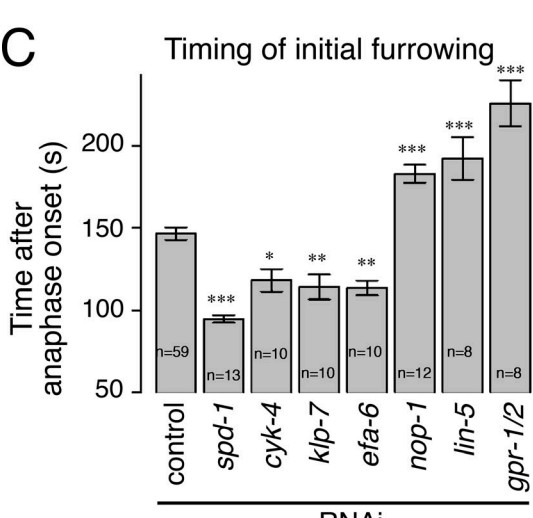

embryos, we could detect 39 cases of internalization, 30 of which were followed by local cortical relaxation. This was reminiscent of the cortical relaxation that is observed after laser ablation (Mayer et al., 2010; Fig. S1). These observations suggest that the removal of myosin II from the cortex leads to a local reduction of the cortical tension/contractility.

### Spindle pole positioning during anaphase generates a local minimum of the aster-cortex interaction at the future cleavage site

Trajectories of the myosin II particles overlaid across embryos illustrate that the internalization of myosin occurs less frequently around the equatorial zone (Fig. 1 E). To test whether the geometry of the anaphase spindle accounts for this spatial

pattern of myosin II internalization, we examined the global distribution of the astral microtubules that reach the cell cortex. A local minimum of the microtubule density at the equatorial future cleavage site has previously been demonstrated in fixed and immunostained *C. elegans* one-cell-stage embryos (Dechant and Glotzer, 2003). However, variable results have been reported by live imaging of microtubules within a plane near the cell surface of embryos that were flattened by being pressed onto a coverglass with an overlaid agarose pad (Bouvrais et al., 2018; Motegi et al., 2006; Verbrugghe and White, 2007). To avoid the possible influence of the deformation of the embryos on the microtubule densities (Harris and Gewalt, 1989), we observed the embryos immobilized on coverglass surfaces without overlaying agarose pads. Recording of the midplane instead of

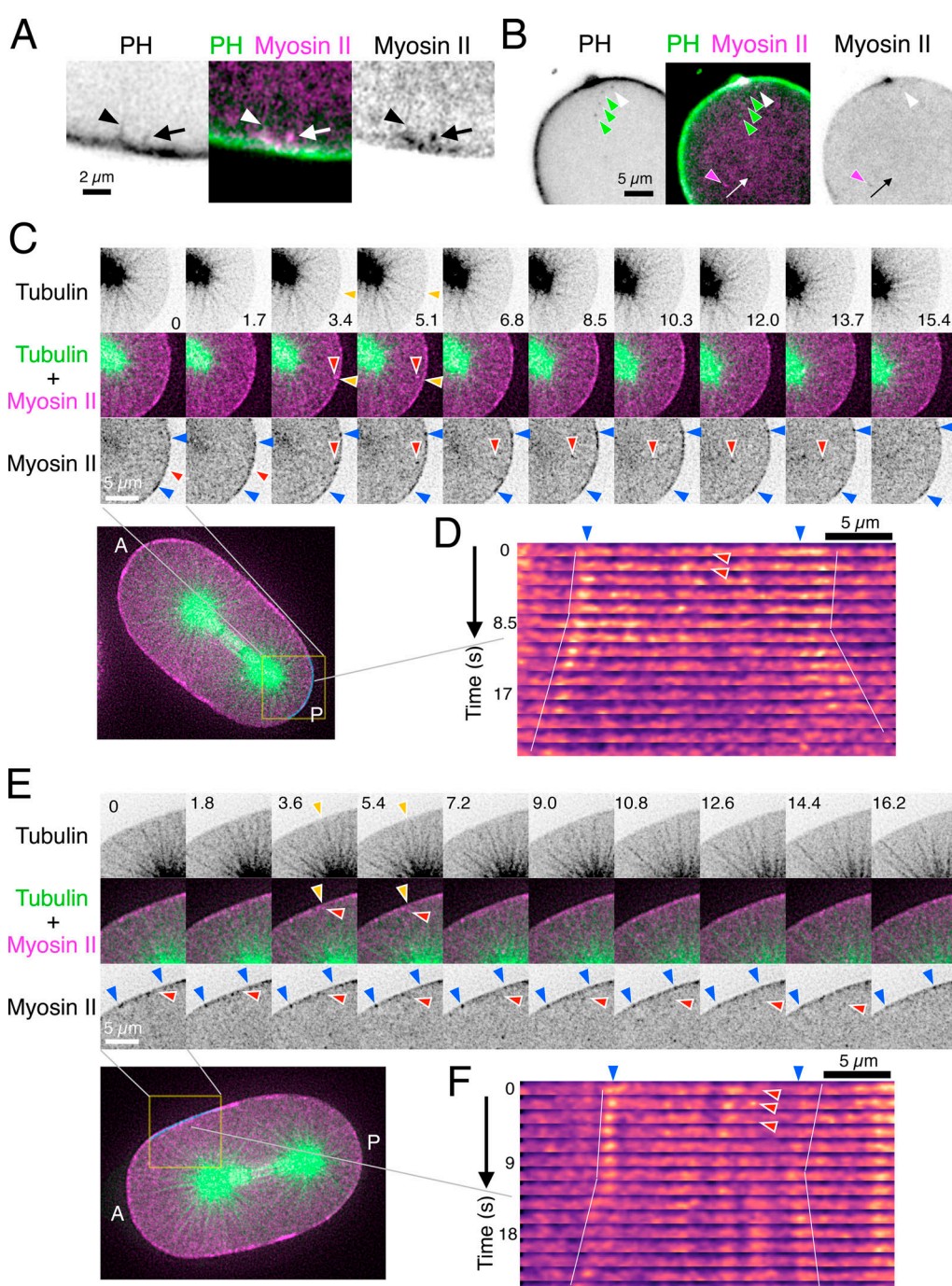

Figure 3. **Local cortical relaxation is observed after removal of myosin II from the cortex. (A and B)** Myosin II-tagRFP-T (magenta) and PH-GFP (green) were simultaneously imaged every 0.77 s with a dual camera spinning disk confocal microscope. **(A)** The invagination of the plasma membrane marked with arrows was led by an internalizing myosin II particle. On the other hand, no myosin particle was found at the tip of another invagination (marked by arrowheads). See Video 5 for the temporal dynamics. **(B)** Some cytoplasmic particles (white arrowhead) showed the signals of both myosin II and the PH domain, while others showed exclusive accumulation of the myosin II (magenta arrowheads) or the PH domain (green arrowheads) signals. See Video 6 for the temporal dynamics. **(C and E)** Embryos expressing myosin II-tagRFP-T (magenta) and tubulin-YFP (green) were imaged by spinning disk confocal microscopy (every 1.7 and 1.8 s in C and E, respectively), and time series of the posterior (C) and the anterior (E) side of two representative embryos are presented. Red arrows indicate the myosin II particles that were detached from the cortex and traveled along a microtubule fiber toward the centrosome. **(D and F)** Line profiles of myosin II on the regions of the cell cortex indicated in C and E, respectively, are presented as kymographs. The particles marked with a red arrowhead were removed from the cortex. Following this, the flanking myosin peaks (blue arrows) were gradually separated from each other, indicating the local relaxation of the cortical actomyosin network.

the flattened surface allowed us to simultaneously observe the spindle poles and the astral microtubules that reach the cell cortex (Fig. 4 A).

To quantify the microtubule density near the cell cortex, we measured the line profile of the YFP-tubulin signal along a curve 1 µm inside the cell boundary (Fig. S2). The height of the peaks that correspond to astral microtubules is very low in comparison with the background signals derived from unpolymerized tubulin, which itself is noisy and variable depending on the location, prohibiting a simple background subtraction (Fig. 4 A). Thus, we calculated the signals that were significantly higher than the local background level (Fig. S2). The average across nine embryos is shown in Fig. 4 B. Reflecting the posterior shift of the spindle after anaphase onset, the density of the microtubules first started to increase at the posterior cortex, creating a global minimum at the anterior tip of the embryo. Then, a local minimum at a site slightly posterior to the equator became clearer around 60 s after anaphase onset and was stably maintained thereafter (Fig. 4 B, green arrows). This spatial pattern is very similar to what was reported by immunostaining (Dechant and Glotzer, 2003), and the spatiotemporal dynamics is consistent with the recent data by live imaging of flattened cell surfaces (Bouvrais et al., 2018).

Astral microtubules in anaphase have been reported to show an exponential length distribution (Dechant and Glotzer, 2003; Redemann et al., 2017). The appearance of the local minimum of the microtubule density at the equatorial cortex described above might be explained simply as a consequence of the distance to the nearby spindle pole. We detected the position of the spindle poles as faint signals of myosin II-GFP (Fig. 4 C, yellow arrowheads). The posterior pole approaching the posterior cortex created a local minimum in the pole-to-cortex distance on that tip of the embryo (Fig. 4 D, 20–60 s), corresponding to the rise of microtubule density on the posterior cortex that precedes the one on the anterior cortex. This was followed by the appearance of a local maximum in the pole-to-cortex distance at the equatorial region (~3 µm posterior from the exact center of the anterior-posterior axis; Fig. 4 D, green arrows), which gradually grew as the anterior spindle pole started to move toward the anterior tip (40–70 s), and predicted the site of cleavage furrow ingression. These observations indicate that the positioning of the spindle poles relative to the cell surface is the primary factor that determines the density of astral microtubules that can influence the cortical activity.

### Promotion of the aster-polar cortex interaction facilitates furrow formation

So far, we have shown that astral microtubules and dynein remove myosin II preferentially from the polar/nonequatorial cortexes and that removal of myosin II from the cortex is frequently associated with local reduction of the cortical tension/contractility. This might provide a mechanism for the polar relaxation, which facilitates the equatorial accumulation of actomyosin. To test this idea, we examined the influence of modifying the geometry of the anaphase spindle relative to the cell surface on the behavior of the cortical actomyosin network.

First, to enhance the aster-cortex interaction, we disrupted the central spindle, which mechanically links the two spindle poles and thus opposes spindle pole separation. We did this by depleting SPD-1, the orthologue of vertebrate PRC1, a highly conserved microtubule-bundling protein crucial for the central spindle formation (Jiang et al., 1998; Mollinari et al., 2002; Verbrugghe and White, 2004; Subramanian and Kapoor, 2012; Fig. 5). In *spd-1(RNAi)* embryos, as expected, the anterior and posterior asters were separated from each other immediately after anaphase onset as a consequence of the rupture of the mitotic spindle (Fig. 5, A and B; Verbrugghe and White, 2004; Lee et al., 2015). This accelerated the approaching of both the poles to the cortex, positioning both of them closer to the cortex than the final levels in the control embryos already within 40 s after anaphase onset (Fig. 5 B). This established a deeper and wider minimum in the pole-to-cortex distance pattern in the anterior cortex, which covered the anterior tip of the embryos, earlier than in control embryos (Fig. 5 A vs. Fig. 4 D, anterior zones in orange). As anticipated, the cytoplasmic transport of the myosin II particles was elevated in the *spd-1(RNAi)* embryos (Fig. 5, C and D; for details, see Materials and methods and Fig. S3) and reached a plateau earlier than in the control embryos. Importantly, despite the disruption of the central spindle, an important source of a positive signal for the contractile ring assembly, the initiation of cleavage furrow formation, was accelerated by 50 s in the *spd-1(RNAi)* embryos (Fig. 2 C).

Slight acceleration of initial furrow formation was also observed when the central spindle was disrupted by depletion of CYK-4, the GTPase-activating protein (GAP) component of centralspindlin, which is crucial for the microtubule-bundling activity as well as for signaling (Mishima et al., 2002; Davies et al., 2015), although in this case the shallow furrow failed to deepen and regressed (Fig. 2 C and Fig. 6 A; Jantsch-Plunger et al., 2000). This indicates that the acceleration of the initial furrow formation by disruption of the central spindle does not depend on the centralspindlin-mediated equatorial stimulation. A similar but milder acceleration of furrow initiation was observed by milder acceleration of the pole-to-pole separation (spindle elongation) by depletion of KLP-7, a microtubule depolymerizer that targets astral microtubules (Grill et al., 2001; Srayko et al., 2005; Rankin and Wordeman, 2010; Han et al., 2015), or by depletion of EFA-6, a negative regulator of the dynein-dependent aster-cell cortex interaction (O'Rourke et al., 2010; Fig. 2 C). These observations further support the role of dynein-mediated myosin II internalization by astral microtubules in the induction of the cleavage furrow.

### Myosin II removal from the polar cortexes triggers a bidirectional cortical flow toward the cell equator

Next, we investigated how dynein-mediated internalization of myosin II controls cleavage furrow formation. To track cortical myosin II dynamics in relation to the geometry of the spindle, we measured the NMY-2::GFP signal along the cell periphery from the anterior tip to the posterior tip at each time point (for details, see Materials and methods and Fig. S4). The average across embryos was displayed as a kymograph representing the dynamics of the distribution of the cortical myosin II (Fig. 5 E).

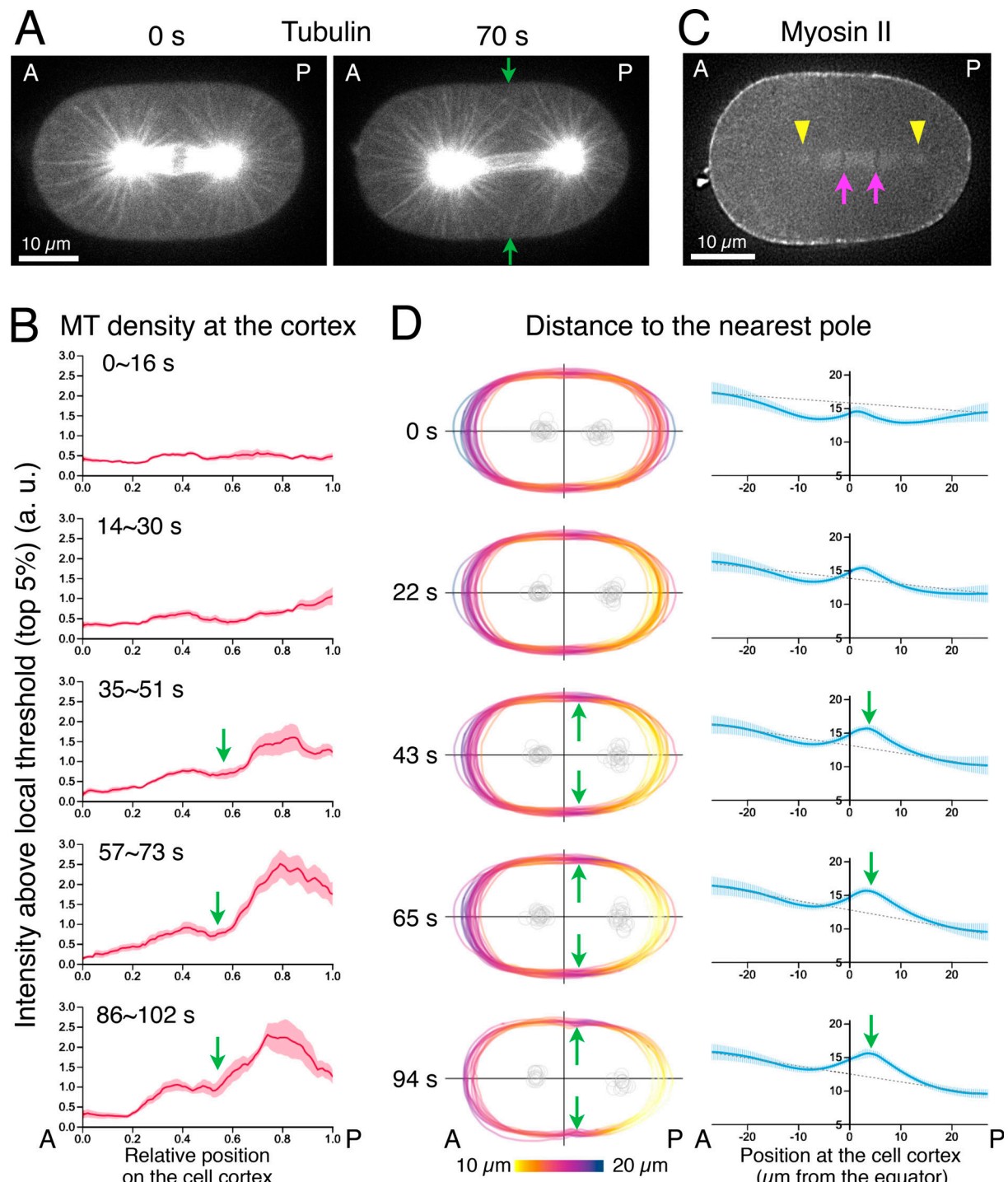

Figure 4. **The position of the spindle poles during anaphase defines a local minimum in the density of astral microtubules that reach the cell cortex at the future cleavage site.** (A and B) Density of microtubules near the cell cortex after anaphase onset. **(A)** Stills from a time-lapse recording of embryos expressing tubulin-YFP. **(B)** The density of the microtubules (MT) along the cell boundary from the anterior (A) tip (0 as the relative position) to the posterior (P) tip (1 as the relative position) in the indicated time window (seconds after anaphase onset). The tubulin/microtubule signals were measured along a line of 1 μm width placed 1.5 μm inside the cell boundary. The intensity above the local background defined by a threshold that corresponds to the top 5% level within a local spatial window (0.2 relative length) was scored as the density of microtubules (see Materials and methods and Fig. S2 for more details) and averaged across the embryos (n = 9, mean ± SEM). **(C and D)** Distance from the cell cortex to the nearest spindle pole. **(C)** The positions of the spindle poles (yellow arrowheads) and the chromosomes (magenta arrows) can be determined in the myosin II-GFP videos. **(D)** The distance from each point on the cell periphery to the nearest spindle pole was measured and presented with color coding (left) and by line plotting (right). The equatorial maximum became prominent after ~40 s and went above a virtual level that linearly links the anterior and posterior tips (gray dotted line). Green arrows indicate the positions of the local minima in the microtubule density (A and B) or the local maxima in the distance to the nearby pole (A and D).

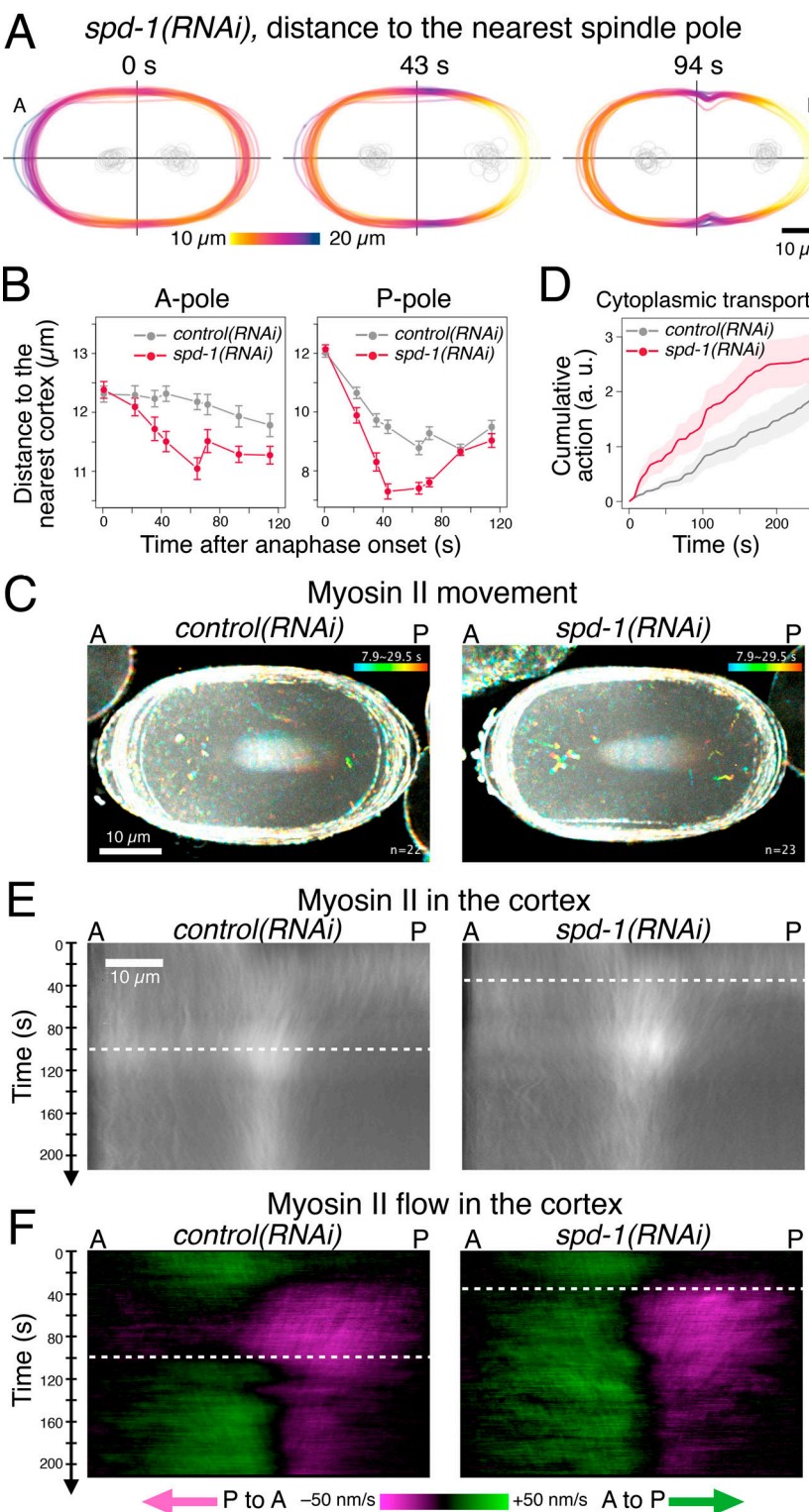

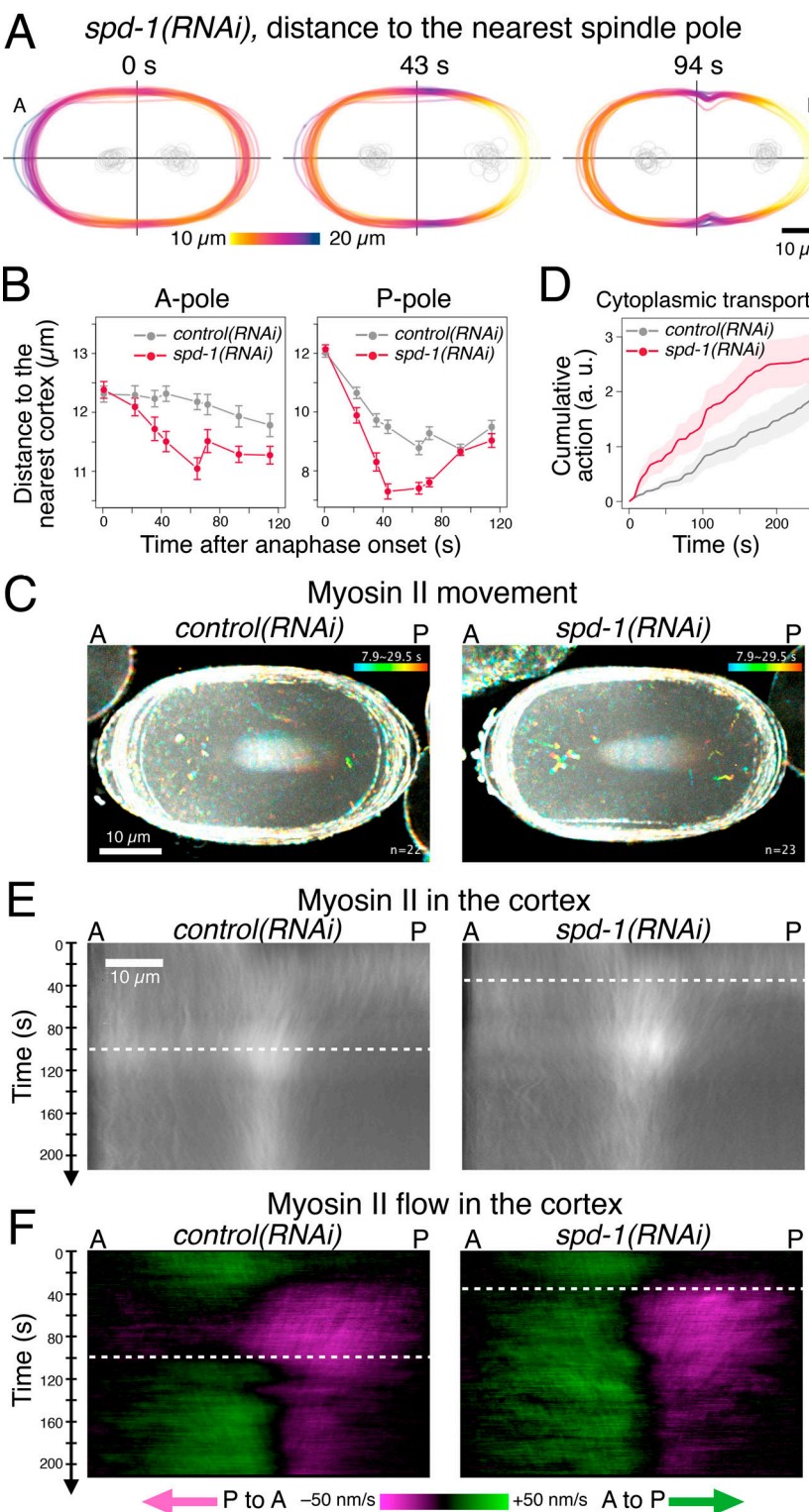

**Figure 5. The spindle pole-cortex geometry regulates the distribution and flow of cortical myosin II. (A and B)** Distance between the spindle poles and the cell cortex. **(A)** The distance from each point on the cell periphery to the nearest spindle pole in embryos depleted of the key microtubule-bundler SPD-1/PRC1 (*spd-1(RNAi)*) was measured and is presented in the same way as the control embryos shown in Fig. 4 D. **(B)** Distance from the anterior pole (A-pole) or the posterior pole (P-pole) to the nearest cortex. Rupture of the central spindle by *spd-1(RNAi)* accelerated the spindle's pole-to-pole elongation, and thus, the decrease in pole-to-cortex distance. **(C and D)** Effect of depletion of SPD-1 on the cytoplasmic myosin II particles. **(C)** Temporal color-coded trajectories of NMY-2::GFP from control (*n* = 22) or *spd-1(RNAi)* (*n* = 23) embryos were overlaid by maximum projection. **(D)** Cytoplasmic myosin II particles were tracked by automation, and the activity of the cytoplasmic transport toward the centrosomes is presented as cumulative sums after anaphase onset (see Materials and methods and Fig. S3 for more details; mean ± SEM, *n* = 22 and 23, respectively). **(E and F)** Density (E) and flow (F) of myosin II in the cell cortex. **(E)** Cortical myosin II (NMY-2::GFP) was quantified along the cell periphery from the anterior tip to the posterior tip and normalized with the local background and cytoplasmic levels. Average across the embryos from two sets of recordings (22 control and 23 *spd-1(RNAi)* embryos for 0 to 140 s p.a.o. and 56 control and 19 *spd-1(RNAi)* embryos for 70 to 210 s p.a.o.) were merged and are presented as a kymograph for each condition (see Materials and methods and Fig. S4 for more details). **(F)** The flow of myosin II along the cell periphery was computed by one-dimensional particle image velocimetry and is presented as a kymograph. The green signal indicates flow toward the posterior and the magenta toward the anterior. The timing of the transition of the cortical flow into the bidirectional mode (white dashed line) is accelerated in the embryos depleted of SPD-1, in which the spindle asters are positioned closer to the cortex.

Using the same data, the cortical flow along the cell periphery was computed in individual embryos by one-dimensional particle image velocimetry (PIV), which gave comparable values to the anterior-posterior component of the flow velocity obtained by 2D PIV (Reymann et al., 2016; Khaliullin et al., 2018; Sugioka and Bowerman, 2018). The average across embryos was displayed as a kymograph in which the green signal indicates flow from the anterior to the posterior, and the magenta indicates flow from the posterior to the anterior (Fig. 5 F).

In normal embryos, during metaphase, myosin II showed an asymmetric cortical localization slightly enriched at the anterior cortex (Fig. 5 E, control, 0 s), as a remnant of post-fertilization polarization (Munro et al., 2004; Werner et al., 2007). After anaphase onset, the signal in the posterior cortex gradually

increased, and subsequently the global signal reached a uniform distribution at around 20 s p.a.o. (Fig. 5 E, control, 0–20 s). This state was maintained until 40 s p.a.o., when the posterior pole approached within ~10 µm from the posterior cortex, forming a minimum at the posterior pole (Fig. 4 D and Fig. 5 B, P-pole). At this point, the signal in the posterior cortex started to decline (Fig. 5 E, control, ~40 s) and, simultaneously, a flow from the posterior to the anterior appeared, which gradually became stronger and extended toward the anterior cortex until ~100 s p.a.o. (Fig. 5 F, control, 40–100 s, the magenta signal on the right half of the panel). This resulted in a broad and moderate accumulation of myosin II at the equatorial zone (Fig. 5 E, control, ~100 s p.a.o.). By contrast, the signal on the anterior cortex stayed largely constant and without a strong flow (Fig. 5, E and F, control, 40–100 s, the left half of the panels). Meanwhile, the anterior spindle pole started to move slowly toward the anterior tip of the embryo (Fig. 4 D and Fig. 5 B, A-pole). Following this period, around 100 s p.a.o., a drop in the myosin II signal was observed in the anterior cortex (Fig. 5 E), concomitantly with the appearance of a strong flow toward the posterior (Fig. 5 F, control, >100 s, the green signal on the left half of the panel). This resulted in a period of bidirectional flow, in which the flow toward the anterior from the posterior cortex converged with the flow toward the posterior from the anterior cortex at the cell equator. The bidirectional flow gradually sharpened the equatorial zone of myosin II accumulation (Fig. 5 E, control) and resulted in furrow formation around 150 s p.a.o (Fig. 2 C).

As described above, the depletion of SPD-1 accelerated the approach of the centrosomes to the polar cortexes and promoted the dynein-mediated internalization of myosin II. This caused a drastic change in the dynamic behavior of myosin II at the cortex. Following the earlier appearance of the anterior minimum in the pole to cortex distance (Fig. 5, A and B), a drop of myosin II signal at the anterior cortex was observed at 60 s p.a.o. (Fig. 5 E, spd-1(RNAi), 60 s p.a.o.), accompanied by an earlier rise of a stronger posterior-directed flow in the anterior cortex (Fig. 5 F, spd-1(RNAi)). This accelerated entry into the phase of bidirectional flow by ~50 s (Fig. 5 F, dashed white lines) and resulted in earlier and stronger accumulation of myosin II at the equator (Fig. 5 E) together with furrow formation ~60 s earlier than in control embryos (Fig. 2 C). Consistent with this, in the cyk-4(RNAi), efa-6(RNAi) and klp-7(RNAi) embryos, in which the approaching of the spindle poles to the polar cortex is mildly accelerated for different reasons (Grill et al., 2001; Srayko et al., 2005; O'Rourke et al., 2010; Lee et al., 2015), the bidirectional flow (Fig. 6) and the furrow ingression (Fig. 2 C) occurred 20–30 s earlier than in control embryos. These observations are consistent with our hypothesis that asters mediate cortical relaxation by removing myosin II from the nonequatorial cortex, thus prompting flow toward the equator.

## Centralspindlin and NOP-1, two upstream regulators of Rho GTPase, regulate the cortical flow in different ways
A nematode-specific protein, NOP-1, has been reported to be another upstream activator of the RhoA GTPase and to work independently of the centralspindlin-ECT2 pathway (Rose et al.,

1995; Tse et al., 2012; Zhang and Glotzer, 2015; Maniscalco et al., 2020). In contrast to centralspindlin, it shows a diffuse localization throughout the cell with slight enrichment at the entire cortex and strong accumulation to the interphase nuclei (Tse et al., 2012). Consistent with its function as a positive regulator of Rho, depletion of NOP-1 significantly delayed furrow initiation (Fig. 2 C), although all the embryos tested could complete cytokinesis as previously reported (Rose et al., 1995; Tse et al., 2012). Cortical flow, especially in the posterior cortex, was decreased in the nop-1(RNAi) embryos (Fig. 6 B), indicating that efficient cortical flow requires the global activity of Rho, which partially depends on NOP-1.

## Polar relaxation by dynein-mediated removal of cortical myosin II works in parallel with the equatorial stimulation
It has been shown that combinations of the depletion of the dynein regulators LIN-5/NuMA or GPR-1/2/Pins/LGN with defects in the central spindle factors cause synthetic failure of furrow formation (Dechant and Glotzer, 2003; Bringmann et al., 2007; Verbrugghe and White, 2007; Werner et al., 2007; Maton et al., 2015). These observations indicate that dynein and dynein regulators work in the furrow induction in parallel with the central spindle/centralspindlin-dependent pathway. However, it has remained unclear whether dynein and its regulators positively control cortical contractility by equatorial stimulation or negatively by polar relaxation.

Here we have found that the depletion of the dynein regulators LIN-5/NuMA and GPR-1/2/Pins/LGN completely eliminates the dynein-driven internalization of myosin II from the cell cortex (Fig. 2 B and Fig. 7 C). Since our results suggest that the removal of myosin II from the cortex leads to local cortical relaxation, it is likely that these dynein regulators work in polar relaxation rather than in equatorial stimulation. To test this and to clarify the contributions of the dynein-dependent and centralspindlin-dependent pathways to the furrow induction, we assessed the effect of the depletion of GPR-1/2 on the dynamics of cortical myosin II and examined the synthetic effect with the inhibition of the centralspindlin-dependent equatorial stimulation observed in cyk-4(or749) mutant embryos (Fig. 7). The point mutation of this temperature-sensitive allele (designated as cyk-4(GAP) hereafter) lies in the GAP domain of CYK-4 (Canman et al., 2008; Davies et al., 2014) and prevents the equatorial stimulation because the GAP domain plays a role in equatorial stimulation by inactivation of Rac (Canman et al., 2008; Zhuravlev et al., 2017) or by activation of Rho via the RhoGEF ECT-2 (Loria et al., 2012; Tse et al., 2012; Zhang and Glotzer, 2015). Importantly, unlike other methods used to perturb the centralspindlin/central spindle-dependent pathway, such as cyk-4(RNAi) (Fig. 2 C; and Fig. 6, A and B), the cyk-4(GAP) mutation, at the restrictive temperature, achieves this without affecting the formation and maintenance of the central spindle or the localization of centralspindlin to the spindle midzone (Canman et al., 2008).

In the embryos with normal CYK-4 but depleted of GPR-1/2, myosin II showed hyperaccumulation to the posterior cortex during early anaphase (Fig. 7 A, iii, gpr-1/2(RNAi), "density"; and Fig. 7 B, gpr-1/2(RNAi)), confirming the role of dynein-driven

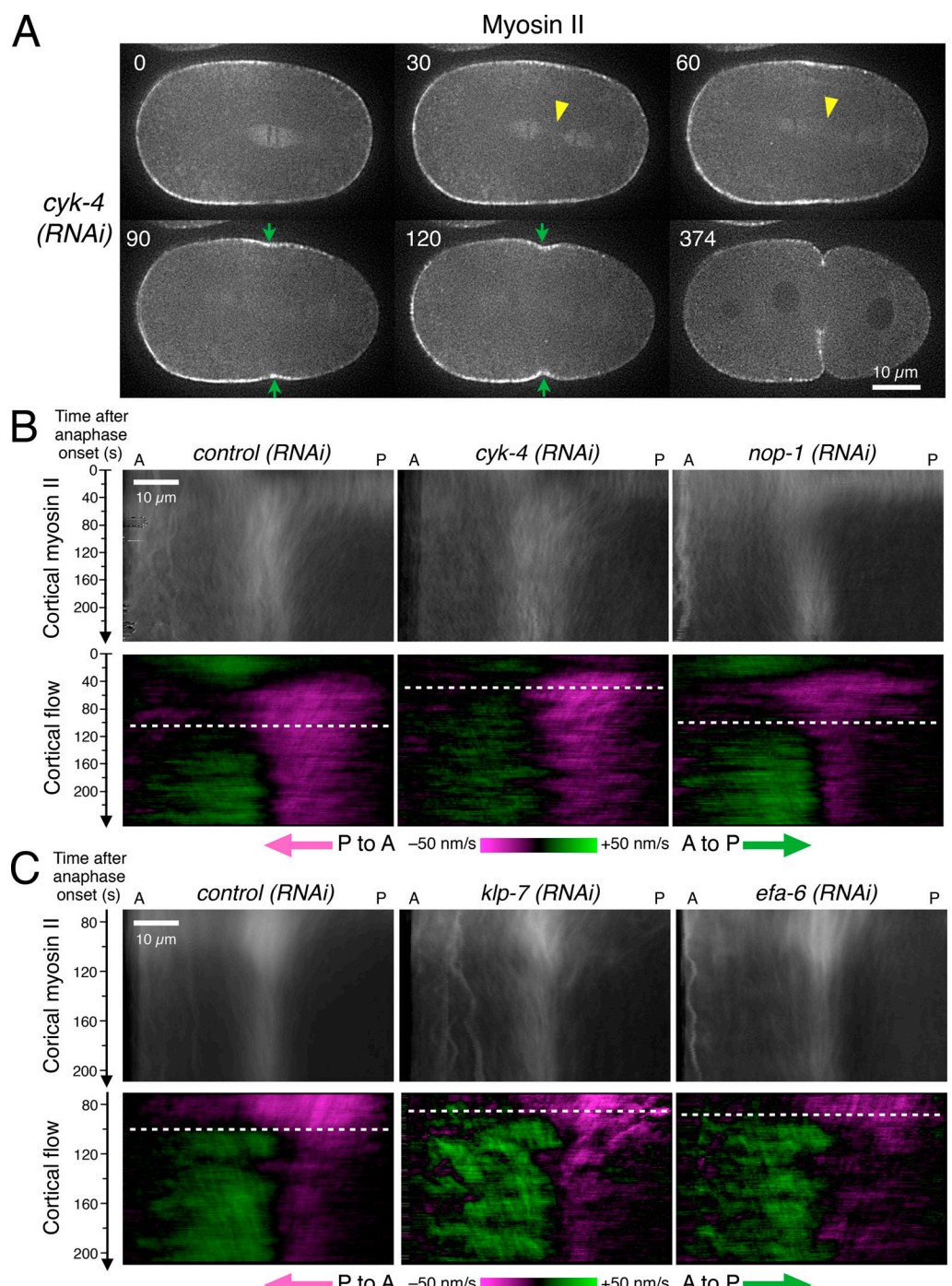

Figure 6. **Promotion of the aster-cortex interaction facilitates the formation of the bidirectional cortical flow.** Further evidence of the acceleration of the formation of the bidirectional cortical flow and the initial furrowing. **(A)** Stills from a video sequence of an embryo expressing myosin II-GFP depleted of CYK-4, the GAP subunit of centralspindlin, in which the central spindle was ruptured (yellow arrowheads) due to defective microtubule bundling, allowing earlier approach of the spindle poles to the polar cortexes. The initial sign of furrowing (green arrows) was detected earlier than in the control embryos (see Fig. 2 C for statistics), but the furrow failed to deepen. **(B and C)** Density (top) and flow (bottom) of the cortical myosin II. **(B)** In addition to the microtubule-bundling activity, centralspindlin has been shown to interact with a RhoGEF, ECT2, and to promote the activity of the actomyosin network. This activity is partially redundant with another upstream activator of ECT2, NOP-1. Consistent with the earlier formation of the furrow (A; Fig. 2 C), depletion of CYK-4 accelerated the timing of the formation of the bidirectional cortical flow (white dashed line). In contrast, in the embryos depleted of NOP-1, the timing of the formation of the bidirectional flow was not affected, although the flow from the posterior to anterior was weaker and the timing of initial furrowing was delayed (Fig. 2 C). **(C)** The aster-cortex interaction is known to be regulated by a microtubule depolymerizer, KLP-7/MCAK, and by a negative regulator of the cortical dynein, EFA-6. In the embryos depleted of either of them, the bidirectional flow formed earlier than in the control embryos, consistent with the earlier initial sign of the furrow formation (Fig. 2 C).

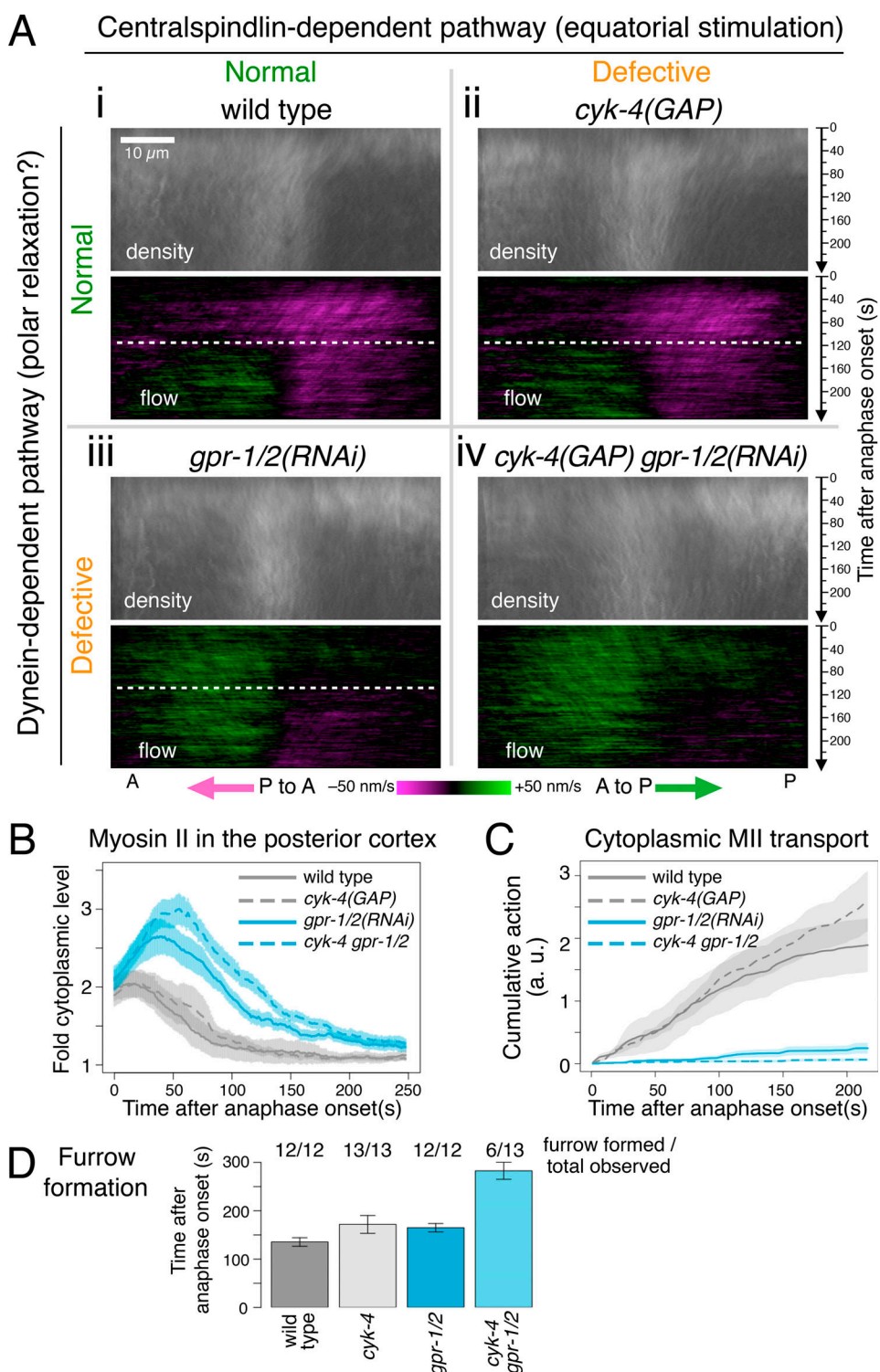

Figure 7. **Polar relaxation by dynein-mediated removal of cortical myosin II induces furrow formation in parallel with equatorial stimulation.** Synthetic effect of the simultaneous inhibition of dynein activity (*gpr-1/2(RNAi)*) and centralspindlin-mediated signaling (*cyk-4(GAP)*) was examined by observing embryos with these mutations in the following combinations: *cyk-4(+) control(RNAi)* (n = 12), *cyk-4(+) gpr-1/2(RNAi)* (n = 12), *cyk-4(GAP) control(RNAi)* (n = 13), and *cyk-4(GAP) grp-1/2(RNAi)* (n = 13). In contrast to *cyk-4(RNAi)*, a point mutation in the GAP domain of CYK-4 (*cyk-4(GAP)*) specifically inactivates the equatorial stimulation via the centralspindlin-ECT2 pathway without affecting the spindle geometry. Since the *cyk-4(GAP)* allele is temperature-sensitive, embryos were imaged at 23°C. **(A)** Density and flow of myosin II in the cell cortex for the indicated strains and conditions were obtained in a similar manner to Fig. 5, E and F. **(B)** Temporal change of the density of myosin II in the posterior cortex within 10 μm from the posterior tip was plotted (mean across embryos ± SEM). **(C)** Cumulative activities of the centrosome-directed cytoplasmic transport of myosin II particles were plotted (mean ± SEM). Inhibition of dynein activity almost completely abolished the myosin transport irrespective of the genotype of *cyk-4* (*gpr-1/2(RNAi)* or *cyk-4(GAP) gpr-1/2(RNAi)*). **(D)** Timing of the initial sign of furrow formation. The numbers on the top indicate the number of embryos that formed the cleavage furrow out of the total number of observed embryos. The

defect in the centralspindlin-dependent equatorial stimulation did not severely affect the patterns of the myosin II distribution and the cortical flow (A, ii, *cyk-4(GAP)* vs. A, i, wild-type). In contrast, inactivation of the cortical dynein activity, which nearly completely eliminated the cytoplasmic transport of myosin II (shown in C), drastically changed the cortical distribution and flow of myosin II (A, iii, *gpr-1/2(RNAi)* vs. A, i, wild-type, and B *gpr-1/2(RNAi)*), although it still allowed a weaker bidirectional flow (A, iii) and furrow formation in the presence of wild-type centralspindlin (D, *gpr-1/2*). Inhibition of both the centralspindlin-dependent pathway and the dynein-dependent pathway almost completely abolished the bidirectional flow (A, iv, *cyk-4(GAP) gpr-1/2(RNAi)*, and B, *cyk-4 gpr-1/2*) and severely affected the furrow formation (D, *cyk-4 gpr-1/2*; see also Fig. S5).

cytoplasmic transport of myosin II, which is almost completely eliminated in *gpr-1/2(RNAi)* embryos (Fig. 7 C, *gpr-1/2(RNAi)*), in the removal of myosin II from the cell cortex (polar relaxation). This was accompanied by a reversal of the global flow that lasts until 120 s p.a.o. (Fig. 7 A, iii, *gpr-1/2(RNAi)*, "flow") and by slightly delayed furrow ingression (Fig. 7 D, *gpr-1/2*). Although the bidirectional flow appeared after 120 s p.a.o., the posterior-to-anterior flow toward the equator in the posterior cortex was much slower than in the wild-type embryos (Fig. 7 A, "flow," iii, *gpr-1/2(RNAi)* vs. i, wild-type). By contrast, in the *cyk-4(GAP)* embryos with a normal level of GPR-1/2, no obvious difference in the cytoplasmic transport of myosin II particles (Fig. 7 C) or in its distribution and flow at the cortex was observed (Fig. 7 A, ii, *cyk-4(GAP)* vs. i, wild-type), indicating a limited role of the centralspindlin-dependent equatorial stimulation in initial formation of the bidirectional cortical flow.

In the embryos that were defective for both the centralspindlin-dependent and dynein-dependent pathways (Fig. 7 A, iv, *cyk-4(GAP) gpr-1/2(RNAi)*), myosin II showed hyperaccumulation at the posterior cortex in a similar manner to that observed in the *gpr-1/2(RNAi)* embryos, with a slight enhancement of the peak height and a delayed clearance (Fig. 7 A, iii, vs. iv, density, and Fig. 7 B). Initially, the entire cortex showed a flow toward the posterior side, where myosin II was enriched, indicating that the contractility per se was not eliminated. Interestingly, however, in these double mutant embryos, the bidirectional flow toward the equator was almost completely abolished (Fig. 7 A, iv, flow), suggesting that the mechanisms for organizing the contractility into properly patterned flows were perturbed. These results indicate that the weak bidirectional cortical flow observed in the *gpr-1/2(RNAi)* embryos in the presence of normal centralspindlin was caused by the centralspindlin-dependent equatorial stimulation. While all the embryos defective only for the centralspindlin-dependent pathway or the dynein-dependent pathway could initiate furrowing, albeit with some delays, about half of the embryos defective for both pathways failed to form a furrow (Fig. 7 D). The rest of the embryos could only form a very shallow furrow, which eventually regressed after a long delay. The depletion of LIN-5 also prevented furrow formation or deepening in the *cyk-4* mutant background (Fig. S5). Taken together, these data indicate that, in the *C. elegans* one-cell-stage embryo, the polar relaxation triggered by dynein-driven myosin II transport contributes to the furrow induction in parallel with the centralspindlin-dependent equatorial stimulation, especially during the early establishment of the bidirectional cortical flow (Fig. 8), although our data do not exclude the possibility that dynein might also contribute to the equatorial stimulation by an unknown mechanism.

## Discussion

Since the first proposal of the polar relaxation by astral microtubules (Wolpert, 1960), its molecular mechanism has remained unclear. Here we present evidence for a fundamental mechanism in which centrosome-directed transport of myosin II along astral microtubules, driven by dynein, reduces the contractility of the polar/nonequatorial cortexes by removing myosin II from them (Fig. 8). The local reduction of cortical contractility triggers a global cortical flow, and the geometry of the mitotic apparatus plays a key role in switching the flow from a unidirectional mode to a bidirectional mode. The bidirectional cortical flow toward the cell equator contributes to the equatorial accumulation of the actomyosin network and to the cleavage furrow formation (DeBiasio et al., 1996; Werner et al., 2007; Zhou and Wang, 2008; Salbreux et al., 2009; Turlier et al., 2014; Reymann et al., 2016).

In addition to the central spindle, the molecules responsible for equatorial stimulation, such as centralspindlin, can be recruited to the anti-parallel overlaps of equatorial astral microtubules (D'Avino et al., 2006; Nishimura and Yonemura, 2006; Argiros et al., 2012; Nguyen et al., 2014; Su et al., 2014; Uehara et al., 2016). Association of the equatorial pools of centralspindlin with the plasma membrane, which is enhanced by its oligomerization, regulated by Aurora B kinase and 14–3–3 proteins (Hutterer et al., 2009; Douglas et al., 2010), has been shown to promote RhoA activation and the tethering of the plasma membrane at the midbody (Lekomtsev et al., 2012; Basant et al., 2015). Based on these, the accelerated furrowing that follows the rupture of the central spindle by SPD-1/PRC1 depletion might be

### Polar relaxation by dynein-driven transport of myosin II

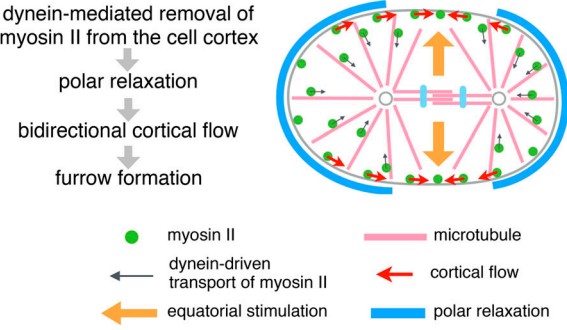

Figure 8. **A model of furrow induction by polar relaxation.** Schematic showing how bidirectional cortical flow triggered by dynein-dependent removal of myosin II from the polar cortexes leads to cleavage furrow formation. In the one-cell stage *C. elegans* embryo, this pathway provides a mechanism crucial for the initial induction of the cleavage furrow before the equatorial stimulation from the central spindle becomes effective.

explained by an earlier release of centralspindlin from the central spindle onto the equatorial asters and the plasma membrane (Adriaans et al., 2019). However, the accelerated induction of a bidirectional cortical flow and furrow initiation by depletion of the CYK-4 GAP subunit (Fig. 2 C and Fig. 6 B) is difficult to explain by this scenario. Our observations rather indicate that the promotion of the aster-cortex interaction can facilitate the furrow formation via the bidirectional cortical flow independently of the equatorial stimulation by central-spindlin; however, we cannot exclude the possibility that a positive signal is released from the central spindle upon its rupture (Baruni et al., 2008; Adriaans et al., 2019) and delivered to the cortex via astral microtubules by an unknown motor protein (Atilgan et al., 2012) earlier than normal.

The equatorial microtubule structures that recruit central-spindlin during natural cytokinesis are organized in an anti-parallel manner and placed lateral to the plasma membrane. In experimentally induced monopolar cytokinesis, the astral mi-crotubules that asymmetrically recruit the midzone proteins to induce furrowing are also laterally associated with the furrow cortex, although they are often bundled in a parallel configura-tion (Canman et al., 2003; Hu et al., 2008; Shrestha et al., 2012; Kitagawa et al., 2013). Here we have shown that the astral mi-crotubules that contact the cell cortex with their plus ends can induce local cortical relaxation by removing myosin II from the cortex. It has also been reported that precise microtubule po-lymerization/depolymerization dynamics does not play a crucial role in furrow induction (Strickland et al., 2005). Considering these observations, we propose a rule of thumb for the action of microtubules on the nearby cell cortex: microtubules laterally associated with the cortex, which are often in an anti-parallel configuration, promote contractility, whereas microtubules pointing toward the cortex relax contractility. It remains unclear whether and how unbundled microtubules approaching the cor-tex at a shallow angle, which are observed at the cell equator in early anaphase, control cortical contractility.

How myosin II is linked to dynein is currently unclear. Whether and how regulators of the cortical pulling forces (Kiyomitsu and Cheeseman, 2013; Kotak et al., 2014; Schmidt et al., 2017; Fielmich et al., 2018; Rodriguez-Garcia et al., 2018; Sugioka et al., 2018) and cortical asymmetry (Chartier et al., 2011; Pacquelet et al., 2015; Sugioka and Bowerman, 2018), as well as membrane additions (Gudejko et al., 2012) or in-vaginations/endocytosis (Tse et al., 2011; Redemann et al., 2010), are involved in dynein-driven transport of myosin II and regulation of other cortical activities (and vice versa) will be important future questions. Although myosin II showed hyper-accumulation to the posterior cortex in the absence of the dynein activities, it was gradually cleared off, albeit with a significant delay, indicating the presence of other mecha-nisms for clearing myosin II from the cortex outside of the cleavage furrow. Another interesting question will be how the dynein-driven removal of myosin II cooperates with or is regulated by mitotic kinases and phosphatases or GTPases that control the contractile actomyosin networks, such as Aurora A, which accumulates at the centrosomes and breaks the symmetry of the cell cortex during zygotic polarity

establishment and cytokinesis (Zhao et al., 2019; Mangal et al., 2018; Klinkert et al., 2019).

Based on the model we propose, we can say that compared with a mechanism that depends on a diffusive signal, direct remodeling of the actomyosin network through a physical in-teraction with microtubules has the advantage of keeping the effect precisely localized at a long distance. We speculate that the dynein-driven myosin II transport mechanism revealed in this work may have a broader biological role in controlling cell shape in other cell/tissue contexts, such as in cell migration and epi-thelial morphogenesis, since both dynein and myosin II are universally expressed in metazoan cells.

## Materials and methods

### *C. elegans* strains and culture conditions
The *C. elegans* strains used in this study are listed in Table S1 and were maintained at 20°C, except for those containing the temperature-sensitive *cyk-4(or749)* allele, which were main-tained at 15°C. LP162 strain (*nmy-2(cp13)*; Dickinson et al., 2013) was crossed with OD56 (*ltIs37*; Essex et al., 2009) and EU1404 (*cyk-4(or749)*; Canman et al., 2008) strains to create QM160 (*nmy-2(cp13[nmy-2::gfp + LoxP]) I; ltIs37[(pAA64) pie-1p::mCherry::his-58] IV*) and QM196 (*nmy-2(cp13[nmy-2::gfp + LoxP]) I; cyk-4(or749) III*) strains, respectively. SWG008 strain (*nmy-2(ges6[nmy-2::tagRFP-T + unc-119(+)]) I*) was crossed with TH65 (*ddIs15[pie-1p::tba-2::YFP]*) and OD58 (*ltIs38[pie-1p::GFP::PH (PLC1delta1)]*) to create QM168 (*nmy-2(ges6[nmy-2::tagRFP-T]) I; ddIs15[pie-1p::tba-2::YFP]*) and QM169 (*nmy-2(ges6[nmy-2::tagRFP-T]) I; ltIs38[pie-1p::GFP::PH (PLC1delta1)]*) strains, respectively. The *gpr-1/2* RNAi clone was from the Vidal Library (Rual et al., 2004; Source BioScience). The *spd-1* RNAi clone was previously described (Lee et al., 2015). The other RNAi clones were from the Ahringer Library (Kamath et al., 2003; Source BioScience). For RNAi in the non–temperature-sensitive strains, L4-stage larvae were placed on the RNAi plates and incubated at 20°C for 48 h before dis-section to obtain embryos. For RNAi in the temperature-sensitive strains, L4 worms were plated and incubated at 16°C for 72 h.

### Live microscopy
For fluorescence live imaging, *C. elegans* embryos were immo-bilized on the coverglass surface of a 35-mm glass-bottom dish (Fluorodish, FD35, World Precision) and cultured in a drop of an osmolality-controlled medium. Before experiments, the center of the coverglass was coated with 20 µl of 0.1 mg/ml poly-L-lysine (molecular weight >300,000, Sigma-Aldrich, P1524) for 1 h. After a quick wash with water, a circle was drawn with a PAP pen (Sigma-Aldrich, Z377821) around the poly-L-lysine–coated area, leaving a hydrophilic surface in the center (~10 mm diameter). A piece of filter paper cut into a donut shape of 32-mm diameter with an 18-mm-diameter hole was placed onto the bottom of the culture dish and made wet with water. Gravid hermaphrodite animals were dissected in 1 µl of dissection buffer (90 mM sucrose, 50 mM EGTA, 5 mM $MgCl_2$, 50 mM potassium acetate, 50 mM Pipes-NaOH, pH 7) dropped at the center of the poly-L-lysine–coated area of the coverglass,

immediately followed by addition of 100 µl of an isotonic medium based on Leibovitz's L-15 medium (Gibco, 21083–027) supplemented with 10% (vol/vol) fetal bovine serum, 35 mM sucrose, 100 U/ml penicillin, and 100 µg/ml streptomycin (Edgar, 1995; Christensen et al., 2002). The sucrose concentration was optimized for the viability and normal embryonic divisions upon eggshell permeabilization by *perm-1* RNAi (Carvalho et al., 2011). To remove unattached embryos and the corpses of the mothers, the medium was replaced with another 100 µl of the same medium. The culture dish was then covered with a 50-mm-diameter coverglass and mounted on the stage of an Andor Revolution XD spinning disk confocal microscopy system based on a Nikon Eclipse Ti inverted microscope equipped with a Nikon CFI Apochromat Lambda S 60×/1.40 NA oil-immersion objective lens, a spinning-disk unit (Yokogawa CSU-X1), and an Andor iXon Ultra EM-CCD camera. Images were acquired using Andor IQ3 software. Fluorophores were excited by laser lines at wavelengths of 488 nm for GFP or 561 nm for mCherry. The temperature of the sample, which was monitored by a FLIR One thermal imaging camera (FLIR Systems, Inc.), was maintained at 21°C (Figs. 1, 2, 3, 4, 5, and 6) or at 23°C (Fig. 7) by circulating cold air within an environmental chamber with a side panel removed. Embryos were staged by differential interference contrast (DIC) observation, and the progress of mitosis was monitored every 5 s by observing the state of chromosomes with histone-mCherry (QM160 strain in Figs. 5 and 6) or as dark zones in NMY-2::GFP (QM168 and QM169 strains in Figs. 3 and 4 and QM196 strain in Fig. 7). For fast recording of NMY-2::GFP, image acquisition was started immediately after anaphase onset (Fig. 4; Fig. 5; Fig. 6, A and B; and Fig. 7) or 70 s later (Fig. 5 and Fig. 6 C) to capture a set of five z-slice images (100 ms exposure, 133 nm/pixel) with 0.5 µm z-steps every 0.83 s (Fig. 1 and Fig. 2), 1.7 s (Fig. 3 C), 1.8 s (Fig. 3 E), 1.6 s (Fig. 4), 0.72 s (Fig. 5 and Fig. 6 C), or 1.25 s (Fig. 5, A and B; and Fig. 7).

Live microscopy of strain QM169 (Fig. 3, A and B) was performed in a similar manner to that indicated above (100 ms exposure, 0.5 µm z-steps every 0.77 s) but using an Andor TuCam system equipped with a Nikon ECLIPSE Ti inverted microscope, a Nikon Plan Apo Lambda 100×/1.45-NA oil-immersion objective lens, a spinning-disk confocal system (CSU-X1; Yokogawa Electric Corporation), two Andor iXon Ultra EMCCD cameras, and a wavelength filter set consisting of a 561-nm single-edge laser-flat dichroic beamsplitter, a 514/30-nm single-band bandpass filter, and a 568-nm ultrasteep longpass edge filter (Semrock). Andor IQ3 software was used for simultaneous image acquisition (69 nm/pixel) of specimens with fluorophores excited by laser lines at wavelengths of 488 nm and 561 nm.

For laser ablation of the anterior cortex in Fig. S1, embryos prepared as above were subsequently mounted on the stage of a 3i Marianas SDC inverted microscope equipped with a Zeiss α Plan Apo 100×/1.46 NA oil-immersion objective lens, a Yokogawa CSU X-10 spinning disk unit, and a 3i Ablate! laser ablation system (532 nm, 1.3 ns pulsed laser). Images were acquired and ablations were performed using 3i SlideBook 6 software with the following parameters for ablation: laser intensity = 100%,

duration = 10 ms, repetitions = 1, raster block size = 3, and rectangle size = 3. GFP was excited with a laser line at a wavelength of 488 nm.

For DIC microscopy in Fig. S5, the embryos were mounted between a coverglass and a 2% (wt/vol) agarose pad in 0.7× egg salt on an FCS2 cooling device (Bioptechs) set at 25°C, and then filmed with an Olympus BX-51 upright microscope equipped with a UPlanSApo 100×/1.4 NA objective, DIC optics, and a CoolSNAP HQ2 CCD camera (Photometrics) controlled by MicroManager (https://www.micro-manager.org/; Edelstein et al., 2014).

### Image analysis

The microscope images were processed and analyzed by custom scripts written with the macro language of Fiji/ImageJ (https://fiji.sc; Schindelin et al., 2012) and R language (https://www.r-project.org/). The four-dimensional images of NMY-2::GFP were deconvolved time frame by time frame with the DeconvolutionLab2 plugin (http://bigwww.epfl.ch/deconvolution/; Sage et al., 2017) using the Richardson-Lucy algorithm with total-variation regularization and a point spread function calculated by the PSF generator plugin (Kirshner et al., 2013) with the Born and Wolf model. The coordinates of the cell periphery were determined frame by frame in the bleach-corrected, average z-projections of nondeconvolved images using the Trainable Weka Segmentation plugin (https://imagej.net/Trainable_Weka_Segmentation; Arganda-Carreras et al., 2017). The positions of the spindle poles were determined by human visual detection of the weak NMY-2 localization to the spindle and the spindle poles in anonymized videos.

The microtubule density near the cell cortex was quantified according to a procedure summarized in Fig. S2. The background outside of the cell was subtracted from the line profile along a curve of 1 µm width, placed 1.5 µm inside the cell boundary (Fig. S2 A) and standardized with the mean intensity (Fig. S2 B). The position along the curve from the anterior pole to the posterior pole was rescaled from 0 to 1. Data from the top and bottom sides of an embryo were pooled for 10 time points (16 s; Fig. S2 C). The intensity data that were ranked within the top 5% of the local area of 0.2 width window were treated as the signal from a microtubule (red dots in Fig. S2 D). After subtracting the threshold levels, the data from nine embryos were averaged (Fig. S2 E).

The velocity of the cytoplasmic NMY-2::GFP particles (Fig. 1 C) was determined by making kymographs of all the trajectories in four anaphase embryos and measuring their gradients. For automated scoring in Fig. 1 E, Fig. 5 D, and Fig. 7 C, cytoplasmic NMY-2::GFP particles were detected in each time frame of the deconvolved and average z-projected videos by finding maxima within the boundary of each embryo using MaximumFinder in Fiji/ImageJ. The particles detected in two consecutive time frames less than 7 pixels apart, which correspond to movement at 1.295 µm/s, were stitched into a trajectory by using a custom R script and overlaid on the original videos to be checked by visual inspection (not shown). The particles that had been trapped on the spindle from metaphase and the false signals derived from the ingressing cleavage furrow were manually omitted.

The distance to the closest cortical point was measured for each time point of a trajectory. The direction of the movement of a trajectory relative to the nearby cortex was assessed by the average rate of the increase of the distance to the closest point on the cortex. For Fig. 1 E, the trajectories from 22 *control(RNAi)* embryos that appeared in six or more time frames (4.32 s) moving away from the cortex were overlaid with temporal color coding.

The activity of cytoplasmic transport of myosin II toward the centrosome was scored according to the procedure shown in Fig. S3. The force to move a particle of the same size and shape at velocity $v$ in a viscous medium is proportional to $v$, and thus the power for the movement is proportional to $v^2$. Although we do not know the exact size and shape of individual particles, as an estimator of the transport activity of a trajectory per frame, we scored $I \times v^2$, where $I$ is the intensity of the fluorescent signal of each particle. To quantify the transport activity toward the spindle pole, the movement of the spindle pole needs to be considered. Although the position of the spindle pole was difficult to determine by automation, that of a chromosome linked to the pole, with the kinetochore microtubules of a constant length (anaphase A is absent in *C. elegans* embryos; Oegema et al., 2001), could be tracked by semi-automation, i.e., by manually specifying the dark spots corresponding to the chromosomes in the first time frame and by repeating identification of a nearby minimum in the next time frame. Thus, the movement of the particles was compensated for the movement of the spindle pole in the A-P direction using the A-P component of the chromosome movement (Fig. S3 A). The $(x, y)$ coordinates of a trajectory (Fig. S3 B) were converted to the centrosome-directed components of the movement, $z$, considering the main axis of the movement and the centrosome/chromosome movement (Fig. S3 C). After smoothing $z$ to $z'$, the $I \times dz'^2$, where $dz'$ is an increment of $z'$ per time point, was calculated for each time point for which $dz' > 0$ (Fig. S3 D) and summed up for all the particles moving away from the cortex (Fig. S3 E, thick lines). For Fig. 5 D and Fig. 7 C, the cumulative sum was calculated (Fig. S3 F). This is the total transport activity from anaphase onset (time 0) and is supposed to represent the overall work executed by dynein for the cytoplasmic transport of the myosin II particles.

For analysis of the temporal change of the cortical density and flow of NMY-2::GFP, first, the region 40 pixels (5.33 μm) inside and outside of the edge of the cell in the average z-projected image was straightened so that the periphery of the embryo that was traced counterclockwise starting from the anterior tip was placed from left to right, and thus the posterior tip ended at the center (Fig. S4). The intensity of the cortical NMY-2::GFP signal, which appeared now as a horizontal line in the middle of the straightened "edge" image, was normalized so that the local background (outside of the cell) is 0 and the local cytoplasmic level is 1. The kymograph of the normalized density at the cell cortex was then generated by reslicing the stack of the time series of the straightened edge images with horizontal lines at intervals of 1 pixel and averaging the five consecutive best focal planes, which corresponds to the peak of the NMY-2::GFP signal at the cell periphery (cortex) of 0.67 μm width. After averaging across multiple embryos, the kymograph was folded

back at the center (at the posterior tip) so that the anterior and posterior tips were placed on the left and right ends, respectively. For Fig. 5, the average kymograph from the dataset obtained from 0 to 140 s p.a.o. and that from 70 to 215 s p.a.o. were merged by linearly changing the blending ratio for the overlapping period (70–140 s p.a.o.) after correction for photobleaching, which is more profound at the cortex than in the cytoplasm due to slower exchange with unbleached molecules.

One-dimensional PIV was performed by a custom R script that compares the one-dimensional distribution pattern of the cortical NMY-2 signal along the cell periphery in a time frame with that of the next time frame. The spatial resolution of the kymograph of the NMY-2 signal was increased fivefold (Fig. 5 and Fig. 6) or 10-fold (Fig. 7) by interpolation. The local velocity was determined as the spatial shift needed to maximize the cross-correlation between the windows of the size of 256 pixels (6.83 μm, Fig. 5 and Fig. 6) or 384 pixels (5.11 μm, Fig. 7) from the two consecutive time points. This was scanned along the cell periphery and repeated through the temporal dimension to make a kymograph of the flow of NMY-2, and the spatial resolution was set back to the original one by averaging. After averaging across multiple embryos, the kymograph of the flow was folded back so that the anterior and posterior tips were placed on the left and right ends, respectively. Positive (anterior to posterior) and negative (posterior to anterior) flow velocities were presented by pseudocoloring with green and magenta, respectively.

### Data accessibility
The original image data used in this study listed in Table S2 are available upon request.

### Online supplemental material
Fig. S1 shows the relaxation of the cell cortex induced by laser ablation. Figs. S2, S3, and S4 illustrate the methods to measure the microtubule density near the cell cortex, the activity of cytoplasmic transport of myosin II, and the density and flow of cortical myosin II, respectively. Fig. S5 shows the synthetic effect of the CYK-4 GAP mutation and the depletion of LIN-5/NuMA or GPR-1/2/Pins/LGN. Videos 1, 2, and 3 show myosin II in wild-type embryos in metaphase, from prometaphase to anaphase and in anaphase, respectively. Videos 4 and 5 show myosin II during anaphase in embryos depleted of DHC-1/dynein heavy chain and LIN-5/NuMA, respectively. Videos 6 and 7 show the dual-color imaging of myosin II and a membrane marker (PH domain). Videos 8, 9, and 10 show the dual-color imaging of myosin II and tubulin/microtubules. Table S1 contains the list of the *C. elegans* strains used in this study. Table S2 lists original image data used in this study.

## Acknowledgments
We thank Behrooz Esmaeili for help in *C. elegans* maintenance, Rob Cross for useful comments on the text, and Computing and Advanced Microscopy Unit, Warwick Medical School, for support on imaging. We thank Stephan Grill (Max Planck Institute of Molecular Cell Biology and Genetics, Dresden, Germany), Julie Canman (Columbia University, New York, NY), Tim Davies

(Durham University, Durham, UK), and the Caenorhabditis Genetics Center (Minneapolis, MN) for providing worm strains.

This work was supported by the Wellcome Trust Senior Investigator Award (WT101885MA) and European Research Council Advanced Grant (ERC-2014-ADG no. 671083) to M.K. Balasubramanian, a Cancer Research UK program grant (C19769/A11985) to M. Mishima, Wellcome-Warwick Quantitative Biomedicine Program (Institutional Strategic Support Fund: 105627/Z/14/Z) Seed Funding to M. Mishima, and a Wellcome-Warwick Quantitative Biomedicine Program Interdisciplinary Summer School scholarship to A. Wray.

The authors declare no competing financial interests.

Author contributions: B. Chapa-y-Lazo, M.K. Balasubramanian, and M. Mishima initiated the project. B. Chapa-y-Lazo, A. Wray, and M. Mishima generated the new *C. elegans* strains. B. Chapa-y-Lazo and M. Hamanaka performed live imaging. M. Mishima performed image analysis. B. Chapa-y-Lazo, M.K. Balasubramanian, and M. Mishima prepared the manuscript. M.K. Balasubramanian and M. Mishima supervised the project.

Submitted: 13 March 2019

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

# Supplemental material

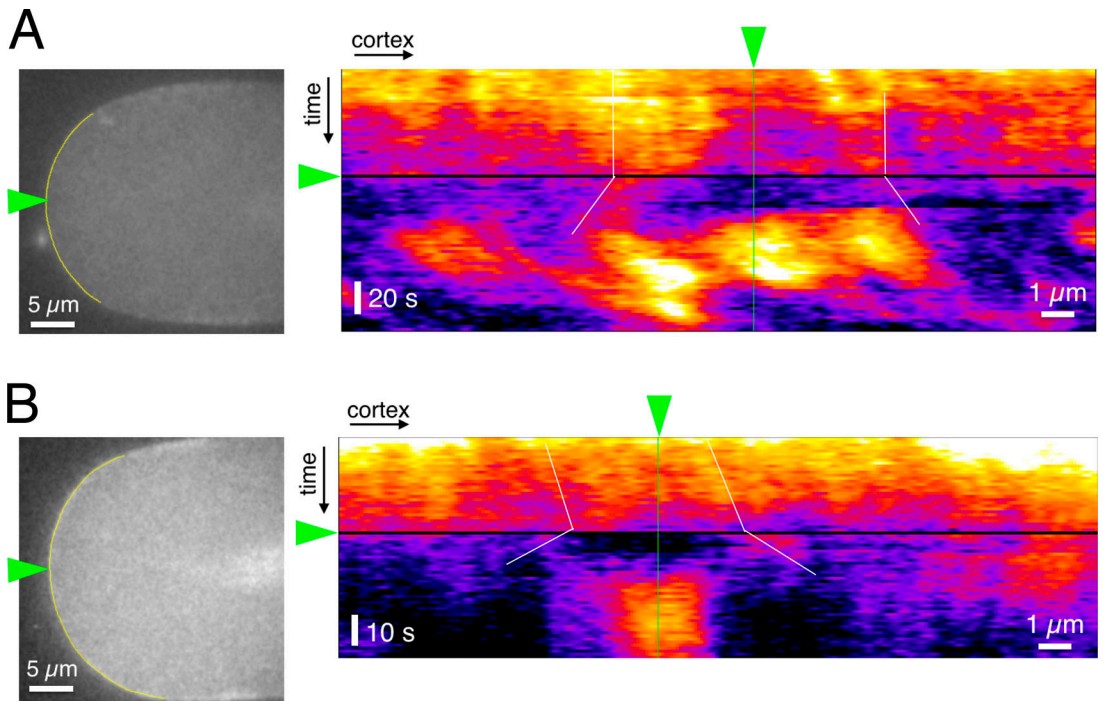

Figure S1. **Laser ablation of the cell cortex causes the local relaxation of the cortical tension/contractility. (A and B)** Two examples of the response of the cortical actomyosin network to laser ablation, monitored by imaging myosin II-GFP. The anterior tip (green arrow, the left panel) was ablated by the illumination with a UV laser. The kymograph of the myosin II distribution along the yellow line is presented on the right. The cortical relaxation was detected immediately after the laser ablation (indicated with white lines) and was followed by hyper-accumulation of myosin II (10–20 s after the ablation), presumably as a process of wound healing.

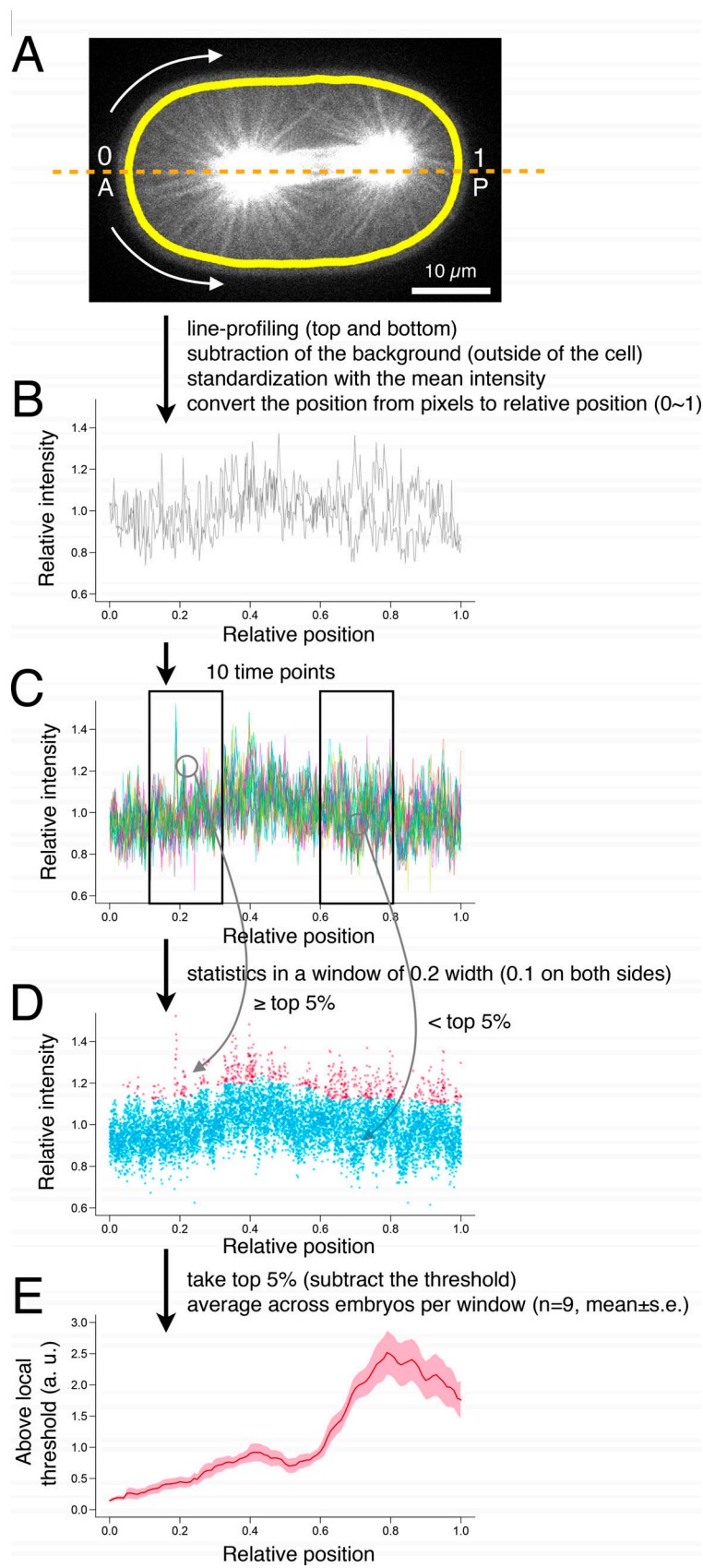

Figure S2. **Measurement of the density of microtubules near the cell cortex.** Related to Fig. 4 B. **(A–D)** The density of the microtubules at a cortical point was determined by line-profiling of tubulin-YFP and by subtracting the local background level. **(E)** The average across nine embryos as shown in Fig. 4 B. See Materials and methods for more details.

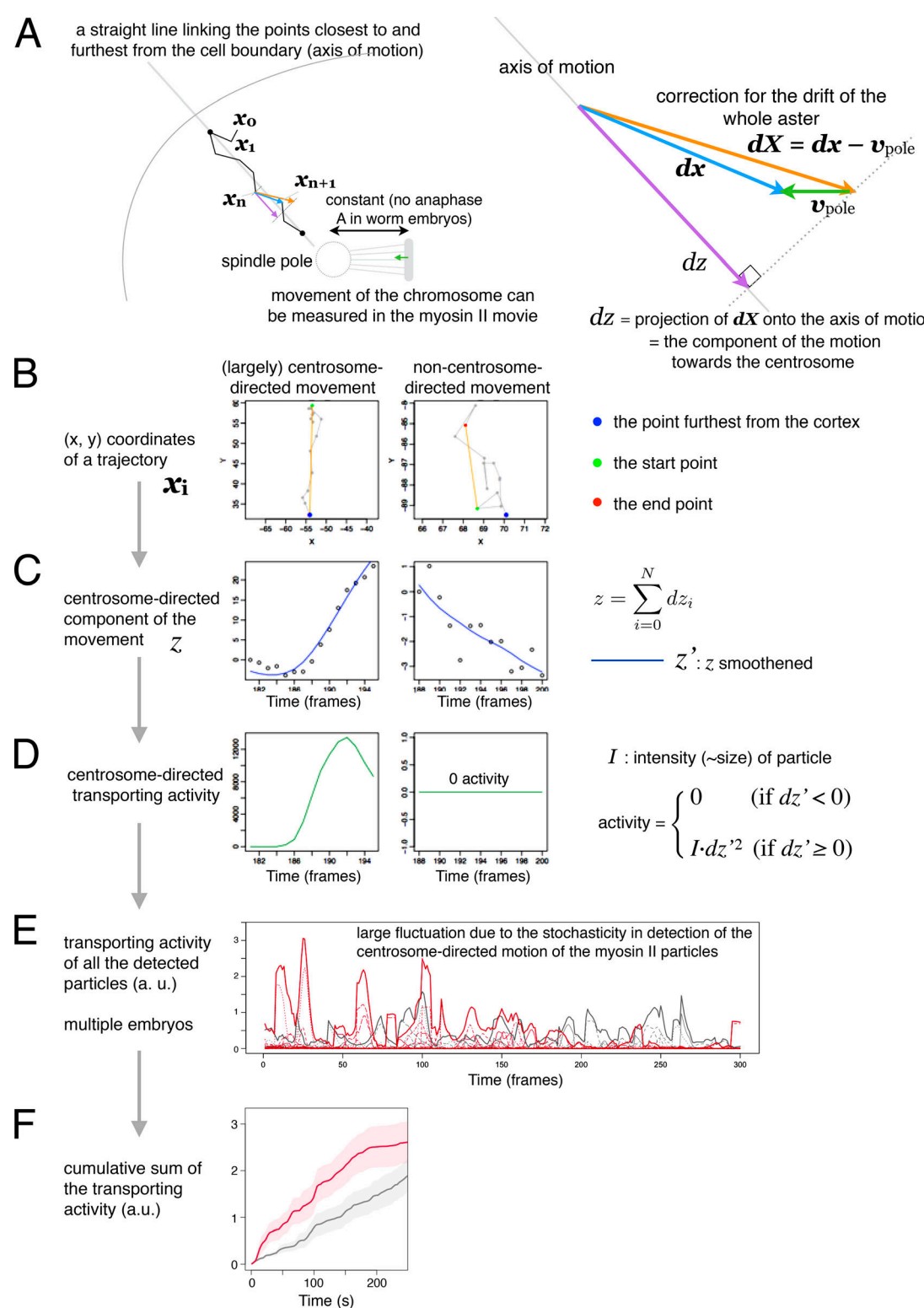

Figure S3. **Scoring the activity of cytoplasmic transport of myosin II.** The activity of the centrosome-directed cytoplasmic transport of myosin II was scored by analyzing the trajectories of the myosin II particles, which need correction for the movement of the whole aster toward the cortex. Only the movement away from the cell cortex was considered. All the particles detected were consolidated across the embryos, and the cumulative sums are presented in the main figures. **(A)** Illustration of the relative geometry of a moving particle, the neaby spindle pole/centrosome and the cell cortex. **(B–D)** Examples of the particles that showed (left) or didn't show (right) a centrosome-directed motion on each step of analysis. **(B)** Raw (x, y) coordinates. **(C)** Centrosome-directed component of the motion and smoothing. **(D)** Calculation of the transport activity. **(E and F)** Summary across multiple embryos before (E) and after (F) cumulative summation through time. **(E)** Thin dotted lines represent the sum of the particle transport activities within individual embryos. Thick lines represent the sum of them across the *spd-1(RNAi)* (red) and the control (gray) embryos, respectively. **(F)** Mean cumulative sums of the transport activities in the *spd-1(RNAi)* (red) and the control (gray) embryos with SEM. See Materials and methods for details.

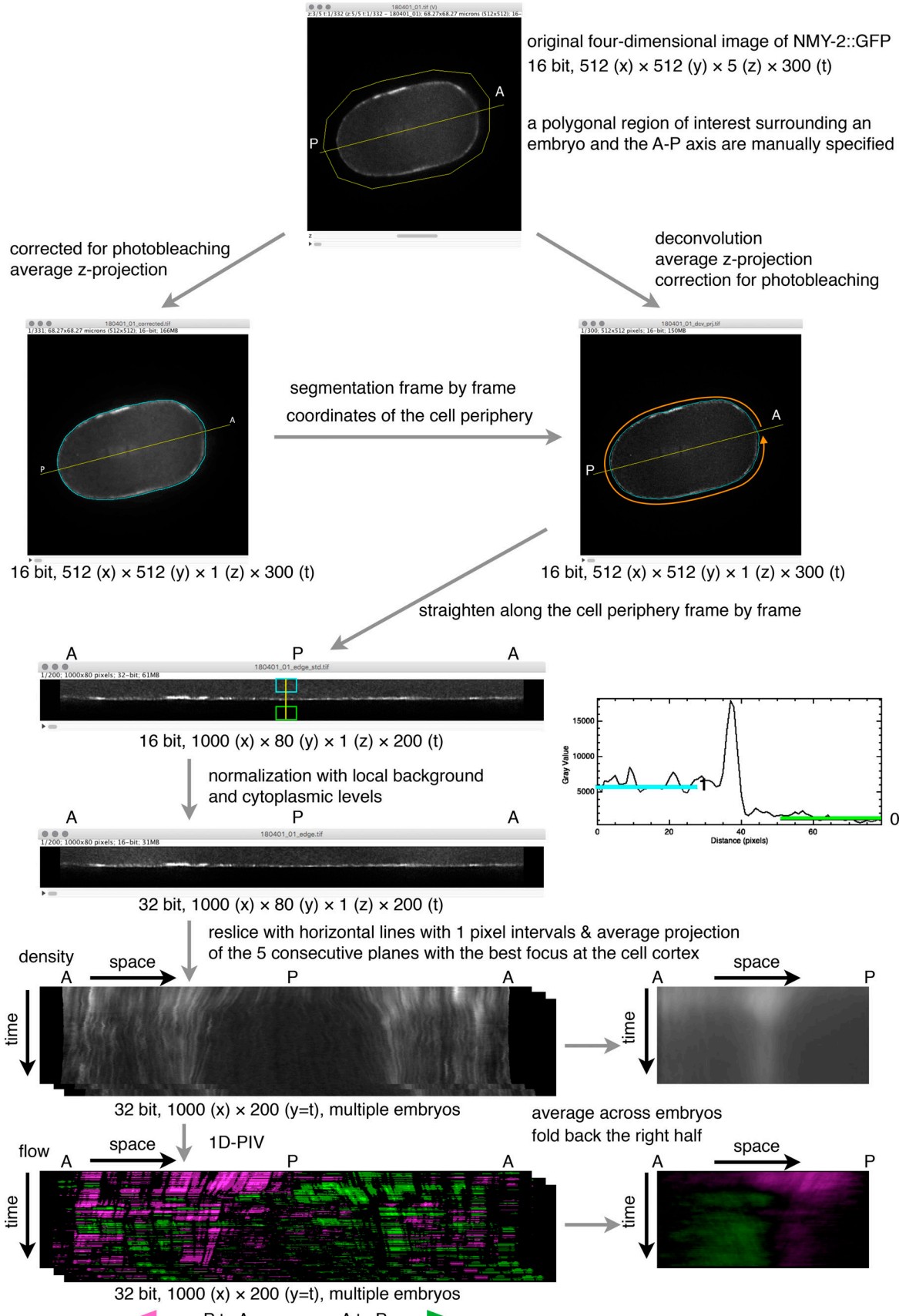

Figure S4.  **Measurement of the density and flow of the cortical myosin II.** A schematic illustration of the procedure of measuring the density and the flow of the cortical myosin II.

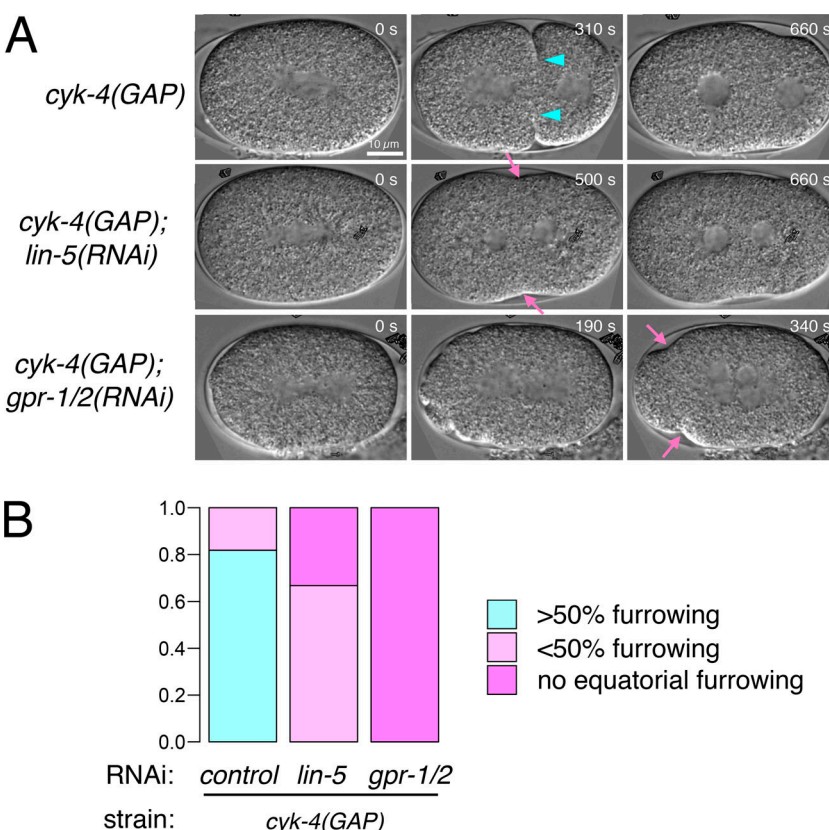

Figure S5. **Synthetic effect of the CYK-4 GAP mutation and depletion of LIN-5/NuMA or GPR-1/2/Pins/LGN. (A)** Stills from the live imaging of embryos of the indicated genotypes by DIC microscopy. While the *cyk-4(or749)* embryos, which failed cytokinesis due to late regression of the cleavage furrow, could form a furrow that deepened beyond 50% (cyan arrowheads in A), the additional depletion of LIN-5 or GPR-1/2 severely prevented this (B), allowing only very shallow furrowing (pink arrows in A).

Video 1. **NMY-2::GFP in the midplane of a *C. elegans* one-cell-stage embryo during metaphase.** The cytoplasmic particles of myosin II only showed random motion. Captured at 0.83 seconds per frame. Playback at 30 frames per second.

Video 2. **NMY-2::GFP in the midplane of a *C. elegans* one-cell-stage embryo from prometaphase to anaphase.** The cytoplasmic particles of myosin II, which showed random motion from prophase to metaphase, gradually disappeared toward the transition to anaphase. After anaphase onset, unidirectional movement of myosin II particles from the cell cortex to the spindle poles was observed. Captured at 1.25 seconds per frame. Playback at 30 frames per second.

Video 3. **NMY-2::GFP in the midplane of a *C. elegans* one-cell-stage embryo during anaphase.** The cytoplasmic particles of myosin II showed unidirectional movement toward the spindle poles. Captured at 0.83 seconds per frame. Playback at 30 frames per second.

Video 4. **NMY-2::GFP in the midplane of a one-cell-stage *dhc-1(RNAi)* embryo in anaphase.** The spindle pole–directed unidirectional motion of the myosin II particles was abolished. Captured at 0.72 seconds per frame. Playback at 30 frames per second.

Video 5. **NMY-2::GFP in the midplane of a one-cell-stage *lin-5(RNAi)* embryo in anaphase.** The spindle pole-directed unidirectional motion of the myosin II particles was abolished. Captured at 0.72 seconds per frame. Playback at 30 frames per second.

Video 6.   **NMY-2::tagRFP-T and PH::GFP in the midplane of a *C. elegans* one-cell-stage embryo during anaphase.** Invagination of the plasma membrane (PH::GFP in green) with or without a myosin II particle (magenta) at the leading tip (Fig. 3 A). Captured at 0.77 seconds per frame. Playback at 5 frames per second.

Video 7.   **NMY-2::tagRFP-T and PH::GFP in the midplane of a *C. elegans* one-cell-stage embryo during anaphase.** Heterogeneity in the colocalization between myosin II (magenta) and the membrane marker (PH::GFP in green) on the cytoplasmic particles (Fig. 3 B). Captured at 0.77 seconds per frame. Playback at 6 frames per second.

Video 8.   **NMY-2::tagRFP-T and tubulin::YFP in the midplane of a *C. elegans* one-cell-stage embryo during anaphase (an example of the videos used for Fig. 3 and Fig. 4).** Images of tubulin/microtubules and myosin II at the midplane of an embryo are shown individually in inverted grayscale (top and bottom, respectively) and as merged images in green and magenta, respectively (middle). Captured at 1.60 seconds per frame. Playback at 30 frames per second.

Video 9.   **NMY-2::tagRFP-T and tubulin::YFP in the midplane of a *C. elegans* one-cell-stage embryo during anaphase.** A myosin II particle (magenta) was internalized from the posterior cortex and moved along a microtubule fiber (green) toward the posterior spindle pole (Fig. 3 C). The sequence is repeated three times. Captured at 1.70 seconds per frame. Playback at 10 frames per second.

Video 10.   **NMY-2::tagRFP-T and tubulin::YFP in the midplane of a *C. elegans* one-cell-stage embryo during anaphase.** A myosin II particle (magenta) was internalized from the anterior cortex and moved along a microtubule fiber (green) toward the anterior spindle pole (Fig. 3 E). The sequence is repeated three times. Captured at 1.80 seconds per frame. Playback at 10 frames per second.

**Two tables are provided online as separate Excel files. Table S1 lists the *C. elegans* strains used in this study. Table S2 lists orignial image data used in this study.**

