## [Peer Review File · The Journal of Cell Biology]

Polar relaxation by dynein-mediated removal of cortical myosin II

Bernardo Chapa-y-Lazo, Motonari Hamanaka, Alexander Wray, Mohan Balasubramanian, and Masanori Mishima

Corresponding Author(s): Masanori Mishima, University of Warwick and Mohan Balasubramanian, University of Warwick

Review Timeline:

Submission Date:	2019-03-13
Editorial Decision:	2019-04-08
Revision Received:	2020-02-03
Editorial Decision:	2020-03-10
Revision Received:	2020-04-09
Accepted:	2020-05-04

Monitoring Editor: William Bement

Scientific Editor: Marie Anne O'Donnell

Transaction Report:

DOI: <https://doi.org/10.1083/jcb.201903080>

April 8, 2019

Re: JCB manuscript #201903080

Dr. Masanori Mishima
University of Warwick
Warwick Medical School
Gibbet Hill Road
Coventry CV4 7AL
United Kingdom

Dear Mohan and Masanori,

Thank you for submitting your manuscript entitled "Polar relaxation by dynein-mediated removal of cortical myosin II". Your manuscript has been assessed by expert reviewers, whose comments are appended below. Although the reviewers express potential interest in this work, significant concerns unfortunately preclude publication of the current version of the manuscript in JCB.

The reviewers are divided as to its suitability for publication by JCB. However, all reviewers agree that the main conclusions are not well supported by the data in the paper. In particular, the reviewers are not convinced that the reported removal of myosin is locally reducing tension. Since this is the major point of the paper, we would need to see additional experiments to bolster this point. At least two possibilities present themselves. One would be ablation experiments in which the tension is quantified following laser severing of the cortex, such as has been performed in other contexts (eg Proc Natl Acad Sci U S A. 2009 Nov 3;106(44):18581-6). Alternatively, it might be possible to detect myosin-removal-dependent differences in cortical tension by immersion of the embryos in hypotonic media. You should also address those issues raised by the reviewers that can be addressed via revisions of the text.

Please let us know if you are able to address the major issues outlined above and wish to submit a revised manuscript to JCB. Note that a substantial amount of additional experimental data likely would be needed to satisfactorily address the concerns of the reviewers. It may be necessary to extend your manuscript to a full Research Article. Our typical timeframe for revisions is three to four months; if submitted within this timeframe, novelty will not be reassessed. We would be open to resubmission at a later date; however, please note that priority and novelty would be reassessed.

If you choose to revise and resubmit your manuscript, please also attend to the following editorial points. Please direct any editorial questions to the journal office.

GENERAL GUIDELINES:

Text limits: Character count for a Report is < 20,000; a full Research Article is < 40,000, not including spaces. Count includes title page, abstract, introduction, results, discussion, acknowledgments, and figure legends. Count does not include materials and methods, references, tables, or supplemental legends.

Figures: A Report may include up to 5 main text figures; a full Research Article may have up to 10 main text figures. To avoid delays in production, figures must be prepared according to the policies

outlined in our Instructions to Authors, under Data Presentation, <http://jcb.rupress.org/site/misc/ifora.xhtml>. All figures in accepted manuscripts will be screened prior to publication.

IMPORTANT: It is JCB policy that if requested, original data images must be made available. Failure to provide original images upon request will result in unavoidable delays in publication. Please ensure that you have access to all original microscopy and blot data images before submitting your revision.

Supplemental information: There are strict limits on the allowable amount of supplemental data. Reports may have up to 3 supplemental figures; a full Research Article may have up to 5 supplemental figures. Up to 10 supplemental videos or flash animations are allowed. A summary of all supplemental material should appear at the end of the Materials and methods section.

If you choose to resubmit, please include a cover letter addressing the reviewers' comments point by point. Please also highlight all changes in the text of the manuscript.

Regardless of how you choose to proceed, we hope that the comments below will prove constructive as your work progresses. We would be happy to discuss them further once you've had a chance to consider the points raised. You can contact the journal office with any questions, cellbio@rockefeller.edu or call (212) 327-8588.

Thank you for thinking of JCB as an appropriate place to publish your work.

Sincerely,

William Bement, Ph.D.
Monitoring Editor

Marie Anne O'Donnell, Ph.D.
Scientific Editor

Journal of Cell Biology

Reviewer #1 (Comments to the Authors (Required)):

Chapa-y-Lazo and colleagues present kinematic and genetic evidence that astral microtubules contacting the cell surface mediate removal of myosin II from the non-equatorial cortex during cytokinesis. They argue that myosin removal reduces local cortical tension, and that because of the relative geometry of cell surface and microtubule array, this tension reduction takes place predominantly outside the equatorial zone, and hence assists with cytokinetic furrow ingression. Their principle evidence that this is so comes from quantitation of cortical myosin flow in wild-type and mutant eggs. They conclude that they have found at least one mechanism for the classic "polar relaxation" proposal for furrow specification, and explain how such a mechanism might work together with stimulation of equatorial contractility by the central spindle.

I enjoyed reading this elegant and straightforward paper. Although none of the specific experiments are definitive in and of themselves, and some repeat observations made by others, together they

make a plausible account for a conceptually simple scenario. As a minor note, I appreciate that these authors used uncompressed eggs for their observations. However I recommend asking the authors to address the following points before acceptance for publication:

First, I am unable to square the kymographs in various figures with the general interpretation that myosin removal reduces cortical tension. The observations reported in Fig. 1d indeed suggest local relaxation. However, in kymographs Figs. 3d,e and 4a, it seems clear that cortical myosin particles either flow in parallel or converge. Parallel tracks imply a sliding sheet, converging tracks imply that the sheet is shrinking in the direction of motion. But polar relaxation predicts locally divergent flow. I can think of several possible explanations, one of which is that the flow is divergent in the orthogonal plane (i.e., along the latitudinal hoops). This could be substantiated through imaging, although it poses its own difficulty, which is that there must be a zone of convergence somewhere around a hoop (which could be why the furrow in these cells rapidly becomes asymmetric). Another possibility is that the actual poles undergo the greatest tension release, in which case it doesn't square with the geometry described in Fig. 3a, which predicts that the first and thus longest effect of the asters should be at mid-latitudes. I'm not sure what all the other possibilities are, but it seems to me that this should be addressed somehow.

Second, an interpretive issue that is related: I urge the authors to consider abandoning the language of equatorial stimulation versus polar relaxation. Although these terms have a long history, I believe they are based on a misleading premise, namely that cortical contractility works like surface tension. If the cortex has a surface tension, then it must be working against internal pressure. I doubt that such an internal pressure exists in most cleaving cells, or at least I am not convinced that the cytokinetic apparatus works against that pressure (as classic treatments, like Wolpert's, Hiramoto's, or White and Borisy, presume).

My argument against such a pressure/tension interpretation is this: numerous physical experiments have been reported in which blunt probes are used to deform cleaving cells, and if the furrow worked against intracellular pressure, then one would see the furrow recoil as a probe deformed the surface. For example, if one pushed on the pole, then if the surface tension interpretation of the cytokinetic furrow was legitimate, the increase in pressure would cause the ingressing furrow to transiently egress. I have done many such experiments on diverse cell types, and cannot recall any instance in which a cytokinetic furrow reversed, even transiently, after the application of a physical deformation.

Indeed the very shape of the *C. elegans* zygote's cleavage furrow argues against a simple tension gradient, which would predict a much more saddle-shaped cleavage furrow than the knife-like cut that is so characteristic of these cells. Instead, it seems to me that for a cell like this one with significant cortical flow, the issue is, why is flow organized the way it is? A similar question has gotten much attention during polarization of the zygote: an initial phase in which transient non-directed contraction generates local ruffles gives way, as the cortex develops a spatial bias in actomyosin recruitment, to globally-directed flow (which likewise converges, rather than the divergence expected from a tearing sheet). The kymographs shown here in Fig. 4a appear to me to support a similar interpretation, i.e., that the *cyk-4/gpr-1/2* combination fails not because contractility fails to emerge, but because the cortex does not flow in an organized spatial pattern. I admit that this is largely a semantic issue, but I think it is time to retire the "polar relaxation versus equatorial stimulation" dichotomy.

Minor comments:

Although the supplemental figures are fairly self-explanatory, they should still have legends.

Fig. 2e would look nicer if the data were arranged shortest bar to longest, with control in the middle.

I'm doubtful about the "simple rule" proposed that relates lateral association of MTs to promotion of contractility, versus inhibition by end-on contacts. First, the nature of the association could be a consequence of dynein engagement inducing catastrophe, as many have demonstrated, in which case both myosin removal and end-on association are consequences of an underlying cause, namely a limited number of cortical pullers. Second, I think it's fairly clear (at least in other cells) that equatorially-directed astral MTs aren't necessarily bundled, nor do they necessarily run along the cortex, and many are stabilized by association with centralspindlin.

Reviewer #2 (Comments to the Authors (Required)):

In this manuscript Chapa-Y-Lazo and coworkers investigate the role of the astral microtubules in polar relaxation during cytokinesis in the *C. elegans* embryo. The authors demonstrate that during anaphase myosin II particles are moving from the cell cortex towards the centrosome and that formation of these particles depends on microtubules and the dynein pathway. The authors propose that myosin II removal results in a local relaxation of the polar cortex which induced an equatorial directed cortical flow which in turn promotes cleavage furrow formation. The proposed model is very interesting and would provide a significant advancement in the field in understanding how the polar cortex relaxes during cytokinesis. However several main points suggested by the authors are currently not well supported by the data and would require major experimental work. Therefore I do not support the publication of the manuscript in JCB.

Two previous publications have demonstrated that membrane invaginations containing myosin II and anillin are internalized during anaphase in *C. elegans* (Tse et al., 2011; Redemann et al., 2010). Similar to the data presented in Fig.1/2 Redemann had also shown that formation of these membrane invaginations is microtubule and *lin-5* & *gpr-1/2* dependent. This makes me wonder whether the myosin particles described here are the same structures published previously. The authors need to clarify this, for example, by imaging PH- and anillin-markers together with myosin II. If the studied myosin II particles are the same as previously described, most data presented in Fig. 1/2 mainly confirms earlier observations (Tse et al., 2011; Redemann et al., 2010). In case the myosin II particles are part of membrane invaginations the authors would also need to revise their model and in order to explain how the internalization of the membrane+cortex particles could result in cortical relaxation.

The second important point proposed by authors is that removal of myosin II results in a local relaxation of the adjacent cortex. The data presented in Fig. 1d supporting a local relaxation after myosin II removal of the cortex is not convincing. In the images presented in Fig. 1d I cannot follow the removed myosin II particle in the cytoplasm and the relaxation of the cortex is also not obvious to me.

Next the authors investigate how the centrosomes-cortex distance influences the formation of the myosin II particles. They use *spd-1* depletion since in *spd-1* RNAi the spindle midzone fails to form and the centrosomes separate faster after anaphase onset (Fig. 3a-b). The authors argue that myosin foci appear earlier in *spd-1* depleted embryos since the asters separate faster after anaphase onset (Fig. 2/3). Since the centrosome-cortex distance is not different in metaphase control and *spd-1*(RNAi) embryos (Fig. 3b) it is surprising to me that intensity of cytoplasmic myosin II particles is already strongly increased at metaphase in *spd-1*(RNAi) embryos (Fig. 2d). This metaphase increase in myosin intensity in *spd-1*(RNAi) background indicates that not the change in centrosome-cortex distance but something else causes this increase. Therefore *spd-1*(RNAi)

seems not a good genetic background to probe the influence of premature centrosome separation on myosin particle formation and cortical flows.

Another essential point of the model is that during centrosome separation more microtubules reach the polar cell cortex and thus increasing number of myosin II particles are removed from the poles. However the authors only measure centrosome-cortex distance but do not directly quantify microtubule distribution. Previous quantifications of microtubule distribution during cytokinesis in *C. elegans* came to contradicting conclusions (Dechant 2003, Motegi 2006, Verbrugghe 2007). To support the proposed model it would be essential to quantify the microtubule number on the cortex in control and different mutant backgrounds over time. In addition, co-localization studies with NMY-2, dynein and microtubules would further support their model that dynein dependent movement of myosin II along microtubules promotes myosin II removal.

Reviewer #3 (Comments to the Authors (Required)):

This manuscript makes a major claim to have identified the molecular basis for aster-dependent polar relaxation in *C. elegans* zygotes, namely through the dynein-mediated removal of cortical myosin II. The existence of an aster-dependent polar relaxation pathway has long been proposed, although it is historically very controversial. The central claim of the manuscript, if it were fully supported by evidence, would certainly be worthy of publication as a JCB report. However, the evidence provided falls short of supporting this central claim and I therefore cannot recommend publication of this manuscript in its current form.

The major problem is that many of the perturbations used give rise to multiple, incompletely-understood defects that are not controlled for or taken into account when reaching the stated conclusions. Alternative, valid interpretations are not entertained and the data end up being correlative rather than conclusive, with regards to the stated conclusions.

This is not to say that high quality data are not presented. Indeed there are many interesting and novel observations that would be worthy of publication if only they were developed a little further in places and more cautiously described and interpreted. The manuscript does show removal of myosin puncta from the polar cortex in a manner that appears to be microtubule- and dynein-dependent, and this removal does correlate with a cortical flow toward the cell equator and the initiation of furrowing. Disrupting dynein function is shown to block myosin removal and to delay furrow formation, while a perturbation that increases the density of astral microtubules (among other things, see below) near the polar cortex accelerates furrow induction. Through monitoring the distribution and flow of myosin at the cortex, a bidirectional flow (from both poles to the equator) is elegantly described that gradually sharpens at the equator prior to furrow ingression. However to claim that this myosin internalization represents the long-sought mechanism of astral relaxation is very premature and quite simply over-interpretation.

Specific points:

1) Transport of myosin on microtubules is a central claim but no microtubules are shown in any of the figures. This should be directly shown.

2) Figure 1d, claims to show myosin signals moving apart after myosin II internalization and it is suggested that this reflects cortical relaxation. It is not obvious how the positions of the red arrows were chosen, and the displayed patch of cortex seems to be as much equatorial as it is polar. A video of this sequence would be helpful, as would additional evidence, e.g. photoconversion of

cortical patches of myosin II and measurements from multiple patches with and without internalization events.

3) *spd-1* RNAi is used as a means to reduce spindle pole-to-cortex distance and increase astral microtubule density at the polar cortex, which it does, but it also totally disrupts the central-spindle (the site at which SPD-1 primarily acts) and, accordingly, this alters the distribution of the centralspindlin-based positive signals that clearly do drive furrow formation. These latter effects on the equatorial stimulation pathway are not controlled for. Indeed, one could argue that the observed differences in myosin movement and furrowing observed upon *spd-1* RNAi could result entirely from changes in the distribution of the centralspindlin-based signals through a combination of: 1) the loss of the ability of the central-spindle to sequester centralspindlin at the cell center, such that all of the signal is now more rapidly able to access the cortex via equatorially-directed astral microtubules, and 2) the increased pole-to-pole distance observed in *spd-1*(RNAi) embryos, e.g. as simulated by Atilgan et al. (2012, PMID: 23001894).

4) The GAP-dead *cyk-4*(*or749ts*) mutant is employed as an alternative means of perturbing the central spindle pathway, but this is complicated by the fact that the precise role of the GAP activity of CYK-4 remains unresolved. It is an over-simplification to simply state that this allele perturbs the equatorial stimulation pathway and that what remains must be the polar-relaxation pathway.

5) Also what of NOP-1, the alternative pathway that acts in parallel with CYK-4 that has been revealed through the analysis of *cyk-4*(*or749ts*) (Tse et al., 2012, PMID: 21737681)? This should be considered and at the very least discussed. Is it involved in this myosin removal pathway?

6) What are the phenotypic consequences of combined perturbations to *spd-1* and dynein? Indeed, this was previously reported by the Mishima group in Lee et al., (2015 PMID: 26088160) where *lin-5* RNAi was shown to suppress the central-spindle defect of *spd-1* (*oj5*) embryos. In that paper it was concluded that dynein-dependent cortical pulling forces contribute to the central spindle breakage observed upon loss of *spd-1* function. How can one separate dynein-dependent myosin transport from dynein-dependent spindle pulling forces, and other functions? This is crucial if a causative link between dynein-dependent myosin internalization and furrowing is to be established.

7) In *lin-5*(RNAi) embryos (Fig 2c, Supp. Video 4), it is claimed that no cytoplasmic movement of NMY-2 was observed, yet clearing of myosin from the polar cortex can be seen, suggesting that myosin internalization is NOT the only mechanism responsible for its delocalization.

Our response to the Reviewers' comments

Original Reviewers' comments

Our response

Chapa-y-Lazo and colleagues present kinematic and genetic evidence that astral microtubules contacting the cell surface mediate removal of myosin II from the non-equatorial cortex during cytokinesis. They argue that myosin removal reduces local cortical tension, and that because of the relative geometry of cell surface and microtubule array, this tension reduction takes place predominantly outside the equatorial zone, and hence assists with cytokinetic furrow ingression. Their principle evidence that this is so comes from quantitation of cortical myosin flow in wild-type and mutant eggs. They conclude that they have found at least one mechanism for the classic "polar relaxation" proposal for furrow specification, and explain how such a mechanism might work together with stimulation of equatorial contractility by the central spindle.

I enjoyed reading this elegant and straightforward paper. Although none of the specific experiments are definitive in and of themselves, and some repeat observations made by others, together they make a plausible account for a conceptually simple scenario. As a minor note, I appreciate that these authors used uncompressed eggs for their observations. However I recommend asking the authors to address the following points before acceptance for publication:

We appreciate this reviewer's positive and constructive comments.

First, I am unable to square the kymographs in various figures with the general interpretation that myosin removal reduces cortical tension. The observations reported in Fig. 1d indeed suggest local relaxation. However, in kymographs Figs. 3d,e and 4a, it seems clear that cortical myosin particles either flow in parallel or converge. Parallel tracks imply a sliding sheet, converging tracks imply that the sheet is shrinking in the direction of motion. But polar relaxation predicts locally divergent flow.

I can think of several possible explanations, one of which is that the flow is divergent in the orthogonal plane (i.e., along the latitudinal hoops). This could be substantiated through imaging, although it poses its own difficulty, which is that there must be a zone of convergence somewhere around a hoop (which could be why the furrow in these cells rapidly becomes asymmetric).

Another possibility is that the actual poles undergo the greatest tension release, in which case it doesn't square with the geometry described in Fig. 3a, which predicts that the first and thus longest effect of the asters should be at mid-latitudes. I'm not sure what all the other possibilities are, but it seems to me that this should be addressed somehow.

The data on the distribution and flow of NMY-2 were calculated along the periphery of the embryos, starting at the anterior pole, passing through the posterior pole and reaching back to the anterior pole (schematics of the image analysis procedure is found in current **Fig. S4**). In current **Figs. 5, 6, and 7**, the data are presented after

folding back at the posterior pole so that the anterior pole is on the left and the posterior pole on the right. The tracks near the posterior pole, which move away from that pole (the right end), indicate the divergent flow, i.e., the relaxation. To clarify this point, in the revised manuscript, we included a kymograph of the posterior cortex of an embryo, which demonstrates the divergent flow from the point of the removal of NMY-2 from the cell cortex (**Fig. 3C** and **3D**).

We regret that our description was not very precise as to the anterior zone of flow emergence while we believe it was reasonable for the posterior one. The cortex-to-pole distance (current **Fig. 4D**) indicates that on the anterior side of the control embryos, shallow minima (orange) appeared not exactly at the anterior tip of the embryos but at the region next to the future furrow. We confirmed that the pattern of microtubule density agrees with this (new **Fig. 4B**): a maximum appears at the region next to the future furrow, not at the exact anterior tip. The pattern of the cortical flow is indeed consistent with these patterns. When the A to P flow appears in the anterior half of the cortex at around 100 s after anaphase onset in the wild type embryos, it appears in the region (~15 μm width) next to the future furrow and no strong flow was detected at the anterior tip (left end on the flow kymograph). This indicates that the flow must be diverging from the interface between these two zones (15~20 μm anterior from the future cleavage plane), which largely corresponds to the place predicated to be under the strongest influence of the astral microtubules both by the distance to the spindle pole (current **Fig. 4D**) and by the actual distribution of microtubules (current **Fig. 4B**). This is also consistent with the distribution of the trajectories of the cytoplasmic myosin II particles (current **Fig. 1E**)

We agree with this reviewer that the cortical flow occurs within the curved two-dimensional surface of the embryos and our approach has a limitation since it cannot assess the flow along the latitudinal hoops (perpendicular to the observed midplane), which might be related to the asymmetric furrow ingression. However, live observation of NMY-2 in 3D volumes at a rate fast enough to capture the cortical flows and cytoplasmic particle movements is highly challenging. We believe that our current data sufficiently demonstrate the average longitudinal (anterior-posterior) dynamics of the cortical myosin II, which is crucial for driving the equatorial accumulation of myosin II that is essential for cytokinesis.

Second, an interpretive issue that is related: I urge the authors to consider abandoning the language of equatorial stimulation versus polar relaxation. Although these terms have a long history, I believe they are based on a misleading premise, namely that cortical contractility works like surface tension. If the cortex has a surface tension, then it must be working against internal pressure. I doubt that such an internal pressure exists in most cleaving cells, or at least I am not convinced that the cytokinetic apparatus works against that pressure (as classic treatments, like Wolpert's, Hiramoto's, or White and Borisy, presume).

My argument against such a pressure/tension interpretation is this: numerous physical experiments have been reported in which blunt probes are used to deform cleaving cells, and if the furrow worked against intracellular pressure, then one would see the furrow recoil as a probe deformed the surface. For example, if one pushed on the pole, then if the surface tension interpretation of the cytokinetic furrow was

legitimate, the increase in pressure would cause the ingressing furrow to transiently egress. I have done many such experiments on diverse cell types, and cannot recall any instance in which a cytokinetic furrow reversed, even transiently, after the application of a physical deformation.

*Indeed the very shape of the *C. elegans* zygote's cleavage furrow argues against a simple tension gradient, which would predict a much more saddle-shaped cleavage furrow than the knife-like cut that is so characteristic of these cells. Instead, it seems to me that for a cell like this one with significant cortical flow, the issue is, why is flow organized the way it is? A similar question has gotten much attention during polarization of the zygote: an initial phase in which transient non-directed contraction generates local ruffles gives way, as the cortex develops a spatial bias in actomyosin recruitment, to globally-directed flow (which likewise converges, rather than the divergence expected from a tearing sheet). The kymographs shown here in Fig. 4a appear to me to support a similar interpretation, i.e., that the *cyk-4/gpr-1/2* combination fails not because contractility fails to emerge, but because the cortex does not flow in an organized spatial pattern.*

I admit that this is largely a semantic issue, but I think it is time to retire the "polar relaxation versus equatorial stimulation" dichotomy.

We agree with this reviewer that there remain many important questions to be answered as to the exact mechanics of the deformation of the cell surface during cytokinesis. We also share a view that the cortical contractility might not simply work like surface tension since the cell cortex is a complex structure consisting of the plasma membrane, which is mechanically a passive element and easily compressible by wrinkling, and the underlining actomyosin networks, which can be more resistant against both the compression and extension and can actively generate mechanical forces. However, since the cell surface is a closed structure that encapsulates the largely incompressible cytoplasm, it would be difficult to describe the mechanics of the shape change of a whole cell during cytokinesis in physics terms without considering the internal pressure. Indeed, when the permeability/osmolarity barrier of the *C. elegans* embryos is disrupted by *perm-1(RNAi)*, furrow formation becomes highly sensitive to the osmolarity of the medium (our unpublished observations).

Nevertheless, the key discovery of our work is that dynein and astral microtubules drive removal of myosin II from the cell cortex, which is associated with the local relaxation of the cortical actin network. This seems to drive the cortical flow, which is divergent (due to relaxation) outside of the equatorial zone but converging (due to contraction) at the furrowing site. This mechanism is in line with the classical theories of polar relaxation and primarily independent of the locally promoted activation of Rho by the centralspindlin-ECT2 pathway as a mechanism for the classical equatorial stimulation. Thus, we think that, in discussing the spatial cues for non-uniform cortical activity, "equatorial stimulation" and "polar relaxation" are still useful concepts. We fully agree with the reviewer's point that the dichotomy on "polar relaxation versus equatorial stimulation" should be retired. Our work clearly demonstrates that the polar relaxation and equatorial stimulation co-exist and cooperate in the cytokinesis of *C. elegans* embryos. We believe that this contributes to ending the ill-defined dichotomy.

The reviewer's interpretation of the combination of *cyk-4(GAP)* and *gpr-1/2(RNAi)* is exactly the same as ours. To make this point clearer, we have rewritten the description of the results of the synthetic genetic study (page 17).

Minor comments:

Although the supplemental figures are fairly self-explanatory, they should still have legends.

We appreciate this comment and have added legends to the supplementary figures.

Fig. 2e would look nicer if the data were arranged shortest bar to longest, with control in the middle.

We modified the bar graph now in **Fig. 2C**, accordingly.

I'm doubtful about the "simple rule" proposed that relates lateral association of MTs to promotion of contractility, versus inhibition by end-on contacts. First, the nature of the association could be a consequence of dynein engagement inducing catastrophe, as many have demonstrated, in which case both myosin removal and end-on association are consequences of an underlying cause, namely a limited number of cortical pullers. Second, I think it's fairly clear (at least in other cells) that equatorially-directed astral MTs aren't necessarily bundled, nor do they necessarily run along the cortex, and many are stabilized by association with centralspindlin.

As to the first point, we agree that the way in which microtubules are associated with the cortex could be influenced by the way in which they interact with the cortical dynein. However, the effect of the interaction between dynein and microtubules would depend on the angle of incidence of microtubules. Microtubules approaching perpendicularly to the cell surface can pull the cortex and remove myosin off more effectively than those running parallel even if they are more sensitive to the contact-induced catastrophe.

As to the second point, as the reviewer points out, not all the equatorially-directed astral microtubules are bundled. At the same time, however, probably, not all such microtubules are responsible for the equatorial activation of Rho. Important are the microtubules that support the accumulation of the upstream regulator ECT2, which in turn relies on centralspindlin for its localization. We have revised the description of our proposal, mentioning its limitations (page 19 to 20).

Reviewer #2:

In this manuscript Chapa-Y-Lazo and coworkers investigate the role of the astral microtubules in polar relaxation during cytokinesis in the C. elegans embryo. The authors demonstrate that during anaphase myosin II particles are moving from the cell cortex towards the centrosome and that formation of these particles depends on microtubules and the dynein pathway. The authors propose that myosin II removal results in a local relaxation of the polar cortex which induced an equatorial directed cortical flow which in turn promotes cleavage furrow formation. The proposed model is very interesting and would provide a significant advancement in the field in

understanding how the polar cortex relaxes during cytokinesis. However several main points suggested by the authors are currently not well supported by the data and would require major experimental work. Therefore I do not support the publication of the manuscript in JCB.

We are pleased to hear that this reviewer found our model “very interesting” and “would provide a significant advancement in the field”. We also appreciate the constructive criticism and we have done our best to address all the points raised:

(Reviewer #2 point 1, Relationship with other phenomena)

Two previous publications have demonstrated that membrane invaginations containing myosin II and anillin are internalized during anaphase in C. elegans (Tse et al., 2011; Redemann et al., 2010). Similar to the data presented in Fig.1/2 Redemann had also shown that formation of these membrane invaginations is microtubule and lin-5 & gpr-1/2 dependent. This makes me wonder whether the myosin particles described here are the same structures published previously. The authors need to clarify this, for example, by imaging PH- and anillin-markers together with myosin II. If the studied myosin II particles are the same as previously described, most data presented in Fig. 1/2 mainly confirms earlier observations (Tse et al., 2011; Redemann et al., 2010). In case the myosin II particles are part of membrane invaginations the authors would also need to revise their model and in order to explain how the internalization of the membrane+cortex particles could result in cortical relaxation.

In the previous work (Tse et al., 2011; Redemann et al., 2010), the invaginations were observed as linear structures marked with PH::GFP or GFP::ANI-1/anillin that were promoted by partial disruption of the cortical actomyosin network, typically by partial depletion of NMY-2/myosin II. Tse et al. (2011) reported that invagination of ANI-1-labeled structures in the control embryos is an infrequent event that retracts within a few seconds. To our knowledge, the behavior of myosin II was not assessed in these papers. The linear appearances of the previously reported invaginations are distinct from the particulate appearances and translational motion of the cytoplasmic NMY-2 signals we describe. On the other hand, the membrane invaginations, the cortical pulling forces and the myosin II internalizations commonly depend on the astral microtubules, dynein and the dynein regulators LIN-5 and GPR-1/2, implying their intimate relationships.

To address the point raised by this reviewer, as suggested, we performed dual-color live microscopy of the embryos co-expressing the GFP-tagged PH domain (PH::GFP) and the myosin II tagged with a red fluorescent protein (NMY-2::tagRFP-T) and those co-expressing GFP::ANI-1 and NMY-2::tagRFP-T in normal embryos.

We detected cytoplasmic particulate signals of the PH domain, some of which colocalized with NMY-2 and sometimes traveled towards the spindle poles similarly to the NMY-2 particles. However, we also observed signals of PH::GFP alone or NMY-2::tagRFP-T alone without clear colocalization. Some invaginations were led by a myosin II particle at their tip, while others didn't show a peak of myosin II at their tip. These observations indicate that some of the myosin II particles might be internalized together with part of the plasma membrane, probably as a vesicle

detached from an invagination. However, this might not be the case for all the NMY-2 particles. These data are now presented in **Fig. 3A** and **3B**.

We also detected centrosome-directed unidirectional movement of the particulate signals of anillin, but they were much less prominent than the brighter cytoplasmic signals (vesicles?) that seemed to be trapped on the anterior aster. This is now mentioned in page 5 and shown in **Fig.1D**.

A recent paper reported the presence of multiple distinct pools of dynein at the cell cortex and the tip of microtubules (Schmidt et al. (2017), PMID: 28739679). The interaction of the astral microtubules with the cell cortex therefore seems to be more complex than had been assumed. As to the invagination and internalization (vesicular) of the plasma membrane, we would also need to consider the roles of the regulation of membrane curvature and the endocytic machinery. We wish to clarify the relationships between the relaxation of the cortical contractility by removal of myosin II and the other phenomena at the cell surface that depend on astral microtubules and dynein in future work.

(Reviewer #2 point 2, Removal of myosin II and relaxation of the local cortex)

The second important point proposed by authors is that removal of myosin II results in a local relaxation of the adjacent cortex. The data presented in Fig. 1d supporting a local relaxation after myosin II removal of the cortex is not convincing. In the images presented in Fig. 1d I cannot follow the removed myosin II particle in the cytoplasm and the relaxation of the cortex is also not obvious to me.

We have performed dual-color live observation of embryos expressing tubulin-YFP and NMY-2::tagRFP-T. Examples are now presented in **Fig. 3 C** to **F**. The process of internalization of a particulate myosin II signal from the cell cortex was continuously monitored throughout its unidirectional movement along astral microtubules towards the spindle pole (**Fig. 3C** and **3E**). The kymographs of the cortical distribution of myosin II around the site of internalization showed a divergent flow away from the site of internalization, indicating local relaxation (**Fig. 3D** and **3F**).

(Reviewer #2 point 3, Timing of appearance of the cytoplasmic myosin II particles)

*Next the authors investigate how the centrosomes-cortex distance influences the formation of the myosin II particles. They use *spd-1* depletion since in *spd-1* RNAi the spindle midzone fails to form and the centrosomes separate faster after anaphase onset (Fig. 3a-b). The authors argue that myosin foci appear earlier in *spd-1* depleted embryos since the asters separate faster after anaphase onset (Fig. 2/3). Since the centrosome-cortex distance is not different in metaphase control and *spd-1*(RNAi) embryos (Fig. 3b) it is surprising to me that intensity of cytoplasmic myosin II particles is already strongly increased at metaphase in *spd-1*(RNAi) embryos (Fig. 2d). This metaphase increase in myosin intensity in *spd-1*(RNAi) background indicates that not the change in centrosome-cortex distance but something else causes this increase. Therefore *spd-1*(RNAi) seems not a good genetic background to probe the influence of premature centrosome separation on myosin particle formation and cortical flows.*

We agree with this reviewer that the increased number of cytoplasmic myosin II

particles before the central spindle rupture due to SPD-1/PRC1 depletion is not consistent with the idea of the removal of cortical myosin II by astral microtubules. We appreciate that this issue was pointed out as we had overlooked it. We took this seriously and addressed it by improving the method of scoring the action of dynein in the cytoplasmic movement of myosin II particles and by testing another method of spindle rupture.

First, we would like to note that it is not very precise to call the first timing of our recordings “metaphase”. We defined the timing of anaphase onset as the first time frame of the NMY-2 recording that was started as early as possible after we detected chromosome segregation in the mCherry::histone images. It might have taken a few seconds between the visual detection of chromosome segregation and the actual start of the GFP::NMY-2 recording. In sum, there might have been a delay from the true timing of anaphase onset and, therefore, it is not very precise to treat the data at time 0 as metaphase. In any case, this itself does not solve the apparent discrepancy pointed out by this reviewer.

The cytoplasmic NMY-2 particles were detected as maxima in each time frame and then stitched into trajectories. In the previous version, for plotting the old Figure 2d, trajectories that appeared in four or more time frames (2.88 s) and moved longer than 3 pixels (0.4 μm) at a velocity faster than 0.3 pixels/frame (55.4 nm/s) were selected. This approach had two problems: 1) the parameter values used for selection of the trajectories were rather arbitrary and 2) the direction of the motility was not considered. This means that the data in the previous version might have been contaminated with the signals that are not related to the action of dynein and astral microtubules, such as particles that were just jiggling in the cytoplasm by Brownian motion, as observed during metaphase (**Fig. 1A** and **Video 1**), and which happened to meet the above selection criteria.

For a more precise analysis, we have omitted the selection based on the arbitrary thresholds and instead calculated a quantity that is expected to more precisely reflect the action of dynein on the NMY-2 movement along astral microtubules. The force necessary to move a particle of a fixed size and shape in a viscous medium should be proportional to the velocity of the motion. Thus, the power executed by dynein to move a particle should be proportional to the square velocity of the movement. In addition, the effect of moving a myosin II particle should be proportional to the number of myosin II molecules in it. Taken together, although we don't know the exact size and shape of the individual particles nor the local viscosity of the cytoplasm, we assumed that (intensity of the particle) \times (velocity)² should serve as an estimator of the activity exerted to transport a myosin II particle. We scored this value for the part of the trajectory that showed the movement away from the cortex. For calculation of the velocity, we considered the direction of the trajectory and the movement of the spindle pole. Please refer to Materials and Methods and **Fig. S3** to see the detailed procedure. To summarize the results, we calculated the cumulative sum of the transporting activity, which is expected to reflect the total amount of work that was executed by dynein on the cytoplasmic movement of myosin II particles after anaphase onset (**Fig. 5D** and **7C**). We could detect elevated myosin II transporting activity in *spd-1(oj5)* embryos as expected (**Fig. 5D**). Importantly, in the first ~10 s, there was no difference between the control

and *spd-1(RNAi)* (**Fig. 5D**). The curve for *spd-1(RNAi)* started to deviate from the control after the central spindle was broken.

(Reviewer #2 point 4, Cortical distribution of the astral microtubules)

Another essential point of the model is that during centrosome separation more microtubules reach the polar cell cortex and thus increasing number of myosin II particles are removed from the poles. However the authors only measure centrosome-cortex distance but do not directly quantify microtubule distribution. Previous quantifications of microtubule distribution during cytokinesis in C. elegans came to contradicting conclusions (Dechant 2003, Motegi 2006, Verbrugghe 2007). To support the proposed model it would be essential to quantify the microtubule number on the cortex in control and different mutant backgrounds over time.

We think that the apparent contradiction between the referred papers is derived from the difference in the methodologies used. An equatorial minimum was detected by observing the midplane of fixed and immunostained embryos (Dechant and Glotzer (2003)) while variable results have been reported by live observation of the cell surface of embryos deformed by an overlaying agar pad (Motegi et al. (2006), Verbrugghe and White (2007) and, more recently, Bouvrais, H. et al. (2018), PMID: 30447992).

We performed live-imaging of the midplane of embryos immobilized without any deformation (current **Fig. 4A** and **Video 7**). Line-profiles along a curve 1 μm inside the cell boundary were analysed. The microtubule signals were quantified as those significantly higher than the local background level (current **Fig. 4B**). Please refer to the Materials and Methods and **Fig. S2** for more details of the procedure.

The MT density profiles for 57~73 s and 86~102 s after anaphase onset (current **Fig. 4B**) was very similar to the wild type pattern in Figure 4E of Dechant and Glotzer (2003), with a minimum located slightly posterior to the precise middle of the cell and a higher maximum on the posterior side than on the anterior side. The equatorial minimum is consistent with the spatial distribution of the cytoplasmic unidirectional motion of the myosin II particles (**Fig. 1E**) and the prediction based on the pole to cortex distance (current **Fig. 4D**), further strengthening our hypothesis. We hope that our original and new data in this revised manuscript help reconcile the discrepancies in the literature.

(Reviewer #2 point 5, Colocalization of dynein on the myosin II particles)

In addition, co-localization studies with NMY-2, dynein and microtubules would further support their model that dynein dependent movement of myosin II along microtubules promotes myosin II removal.

We have demonstrated that the myosin II particles travel along astral microtubules (Reviewer #2 point 2, current **Fig. 3C-E**). We agree that it would further support our model if we could demonstrate co-localization of dynein on the moving myosin II particles. However, no other known motor protein can drive the minus-end-directed motion at the velocity observed for the myosin II particles. We think that the existing RNAi data are reasonably sufficient to assume that dynein is on the moving myosin II particles, and the colocalization would not add much to our main conclusions.

Reviewer #3:

*This manuscript makes a major claim to have identified the molecular basis for aster-dependent polar relaxation in *C. elegans* zygotes, namely through the dynein-mediated removal of cortical myosin II. The existence of an aster-dependent polar relaxation pathway has long been proposed, although it is historically very controversial. The central claim of the manuscript, if it were fully supported by evidence, would certainly be worthy of publication as a JCB report. However, the evidence provided falls short of supporting this central claim and I therefore cannot recommend publication of this manuscript in its current form.*

The major problem is that many of the perturbations used give rise to multiple, incompletely-understood defects that are not controlled for or taken into account when reaching the stated conclusions. Alternative, valid interpretations are not entertained and the data end up being correlative rather than conclusive, with regards to the stated conclusions.

This is not to say that high quality data are not presented. Indeed there are many interesting and novel observations that would be worthy of publication if only they were developed a little further in places and more cautiously described and interpreted. The manuscript does show removal of myosin puncta from the polar cortex in a manner that appears to be microtubule- and dynein-dependent, and this removal does correlate with a cortical flow toward the cell equator and the initiation of furrowing. Disrupting dynein function is shown to block myosin removal and to delay furrow formation, while a perturbation that increases the density of astral microtubules (among other things, see below) near the polar cortex accelerates furrow induction. Through monitoring the distribution and flow of myosin at the cortex, a bidirectional flow (from both poles to the equator) is elegantly described that gradually sharpens at the equator prior to furrow ingression. However to claim that this myosin internalization represents the long-sought mechanism of astral relaxation is very premature and quite simply over-interpretation.

We thank the referee for the valuable comments. At the outset, we wish to state that we are NOT proposing the mechanism we discovered as the only mechanism for astral relaxation. Instead, we are proposing this as one mechanism for it. We are not trying to exclude other possibilities such as negative regulation of Rho by astral microtubules. In the revised version, we modified the text to clarify this point (page 20-21).

Specific points:

1) Transport of myosin on microtubules is a central claim but no microtubules are shown in any of the figures. This should be directly shown.

Dual-color live observation of tubulin::YFP and NMY-2::tagRFP-T was performed and a sequence demonstrating the transport of the myosin II particles along astral microtubules is now presented in **Fig. 3C and E**. We have also analyzed the cortical distribution of the astral microtubules (**Fig. 4B**) and confirmed that a local minimum is formed at the location predicted by the pole-to-cortex distance (**Fig. 4D**) and at the

site of division in good agreement with the polar relaxation. Please also refer to our response to Reviewer #2 point 4.

2) Figure 1d, claims to show myosin signals moving apart after myosin II internalization and it is suggested that this reflects cortical relaxation. It is not obvious how the positions of the red arrows were chosen, and the displayed patch of cortex seems to be as much equatorial as it is polar. A video of this sequence would be helpful,

In the current **Fig. 3 C to F**, we show the myosin II internalization followed by the cortical relaxation alongside the distribution of the astral microtubules. In the previous version, the patches indicated by red arrows were chosen as those that could be visually traceable. In the revised version, a kymograph of the cortical myosin II around the site of internalization is presented. As suggested, we have now included **Videos 8** and **9**, corresponding to **Fig. 3C** and **3E**, respectively.

We apologize that our description was not clear as to the cortical zones where myosin II internalization occurs. The myosin II internalization is not restricted to the cortexes at the anterior or posterior tips of the embryos. It can be observed in wider regions of the cortex. To clarify this, the overlaid trajectories from 22 embryos is now presented as **Fig. 1E**.

as would additional evidence, e.g. photoconversion of cortical patches of myosin II and measurements from multiple patches with and without internalization events.

We appreciate this excellent suggestion. However, we are afraid that generation and characterization of suitable strains and establishment of the conversion conditions would be tedious and would take a long time. We believe that the video sequences presented in **Figure 3** and **Videos 5, 6, 8 and 9** clearly demonstrate the process of internalization of the cortical patches followed by their movement along the astral microtubules towards the spindle pole.

3) spd-1 RNAi is used as a means to reduce spindle pole-to-cortex distance and increase astral microtubule density at the polar cortex, which it does, but it also totally disrupts the central-spindle (the site at which SPD-1 primarily acts) and, accordingly, this alters the distribution of the centralspindlin-based positive signals that clearly do drive furrow formation. These latter effects on the equatorial stimulation pathway are not controlled for. Indeed, one could argue that the observed differences in myosin movement and furrowing observed upon spd-1 RNAi could result entirely from changes in the distribution of the centralspindlin-based signals through a combination of: 1) the loss of the ability of the central-spindle to sequester centralspindlin at the cell center, such that all of the signal is now more rapidly able to access the cortex via equatorially-directed astral microtubules, and 2) the increased pole-to-pole distance observed in spd-1(RNAi) embryos, e.g. as simulated by Atilgan et al. (2012, PMID: 23001894).

The localization of centralspindlin at the furrow tip was previously reported to be enhanced by the disruption of the central spindle (Verbrugghe and White (2004), PMID: 15458647; Zhang and Glotzer (2015), PMID: 26252513). However, it remained unclear whether this cortical localization precedes the furrow formation.

Thus, we performed dual-color spinning-disc live microscopy of the *spd-1(oj5)*

embryos expressing mCherry::tubulin and CYK-4::GFP (**Reviewer Figure 1**). The localization of CYK-4 at the tip of an ingressing furrow was observed as reported (**Reviewer Figure 1A**, blue arrowhead). At the cell surface contacting the coverslip due to the overlaid agarose pad, gradual accumulation of CYK-4 was observed (**Reviewer Figure 1A**, white arrow), which started as fuzzy clouds, coalesced into threads at the overlaps of the astral microtubules from the opposite poles and sharpened into an arc upon furrow ingression. Interestingly, before these cortical CYK-4 accumulations were detected, a cleavage furrow had already formed (top side of the embryo, Reviewer Figure 1A, white arrow at 0 s). At the bottom side of the embryo, a sign of furrow ingression was observed at 40 s (white arrowhead); however, no particular enrichment of CYK-4 signal was detected at this site (**Reviewer Figure 1B**, white arrowhead, viewed from different angles) This suggests that the accelerated furrow formation in the *spd-1(oj5)* embryos can't simply be explained by centralspindlin re-distributed to the cell cortex.

As a complementary approach, we disrupted the central spindle and eliminated the centralspindlin-dependent signaling at the same time by depletion of CYK-4. If the redistribution of centralspindlin to the cell cortex was a pre-requisite for the

accelerated formation of the bidirectional cortical flows observed in the embryos depleted of SPD-1, depletion of CYK-4 would not cause such acceleration. However, upon *cyk-4(RNAi)*, we observed a similar set of phenotypes to those observed in *spd-1(RNAi)* though slightly milder, i.e., accelerated spindle pole separation (**Fig. 6A**), earlier establishment of the bidirectional cortical flows (**Fig. 6B**), and earlier initiation of furrowing (**Fig. 2C**). This clearly indicates that the accelerated establishment of the bidirectional flow upon the rupture of the central spindle doesn't require the centralspindlin-dependent signaling.

Taken together, these new results suggest that the accelerated cortical flow that follows the rupture of the central spindle is difficult to explain by the scenario suggested by Reviewer 3. This is in good accordance with our observation that the early dynamics of the cortical flow was not affected by the *cyk-4(or749)* mutation, which inactivates the positive signaling for the furrow formation (regardless of the molecular mechanisms, see point 4 below for more details) but affects neither the formation nor maintenance of the central spindle (**Fig. 7A i vs ii**). Very strictly speaking, however, there remains the possibility that an unknown signal that is normally sequestered by the central spindle is released by the disruption of the central spindle and delivered to the equatorial cortex by such a mechanism as proposed by Atilgan et al. (2012). We discussed this possibility in Discussion (page 19).

4) The GAP-dead cyk-4(or749ts) mutant is employed as an alternative means of perturbing the central spindle pathway, but this is complicated by the fact that the precise role of the GAP activity of CYK-4 remains unresolved. It is an oversimplification to simply state that this allele perturbs the equatorial stimulation pathway and that what remains must be the polar-relaxation pathway.

It is true that a long-lasting argument has not been settled yet as to the precise molecular function of the GAP domain of CYK-4, namely, "what is the target of the GAP domain of CYK-4?". In one theory, the target is Rac, and local suppression of the Rac activity (which seems to be interfering with various aspects of cytokinesis) promotes cytokinesis (Canman et al. (2008), PMID: 19056985; Nunes Bastos (2012), PMID: 22945935). In another theory, CYK-4 contributes to the local activation of Rho by facilitating the rapid turnover of the GTP/GDP cycle (Miller and Bement (2009), PMID: 19060892) or by allosteric activation of the Rho GEF ECT2 (Zhang and Glotzer (2015), PMID: 26252513). In either case, the CYK-4 GAP domain has some positive roles in the formation and maintenance of the cleavage furrow. Considering its localization, it is highly unlikely that centralspindlin has a direct role in polar relaxation and that the *cyk-4(or749)* mutation interferes with this role. We think it is quite reasonable to assume that this GAP allele perturbs the centralspindlin-dependent pathway for equatorial stimulation.

There remain arguments regarding to what extent the activity of centralspindlin-dependent signaling is reduced in the *cyk-4(or749)* embryos. According to the "inactivation of Rac" theory, the equatorial Rho activity should not primarily be affected. On the other hand, the "(local) activation of Rho" theories would predict a reduced level of the equatorial activation of Rho in the mutant embryos. In either case, however, if dynein and the cortical dynein regulators work for cytokinesis exclusively on the equatorial stimulation, their defects would not strongly prevent

furrow formation in the *cyk-4(or749)* embryos since the remaining intact polar relaxation signaling would be able to induce furrowing. The observed synthetic furrowing defects in **Fig. 7** contradict this and support the role of the dynein-dependent pathway in polar relaxation, although the possibility that it works in both equatorial stimulation and polar relaxation cannot be excluded. We clarified this point by mentioning this possibility in the Results section (page 17-18) and by changing the annotations on Fig. 7A.

*5) Also what of NOP-1, the alternative pathway that acts in parallel with CYK-4 that has been revealed through the analysis of *cyk-4(or749ts)* (Tse et al., 2012, PMID: 21737681)? This should be considered and at the very least discussed. Is it involved in this myosin removal pathway?*

NOP-1 is a poorly conserved, non-essential protein, which localizes both to the nucleus and the cytoplasm with some enrichment at the cortex of the pseudocleavage furrow and the cytokinetic furrow. While the pseudocleavage is abolished in the absence of NOP-1, cytokinesis can be completed although with a slight delay in the timing of furrow initiation. Based on the defects of *nop-1* in pseudofurrowing and the synthetic defect of *cyk-4;nop-1* in furrow formation, NOP-1 has been proposed to be a global activator of ECT2 that acts in parallel with centralspindlin, which acts more locally (Tse et al. 2012; Zhang & Glotzer 2015).

As suggested, we examined the effect of *nop-1(RNAi)* on myosin II dynamics. Consistent with a previous report (Tse 2012), a delay in the timing of furrow initiation was detected (current **Fig. 2C**). Cytoplasmic movement of myosin II particles towards the spindle poles was not abolished by *nop-1(RNAi)*, indicating that NOP-1 is not essential for dynein's action on myosin II (**Reviewer Figure 2**). We observed a slightly diminished cortical flow (new **Fig. 6B**), which might be reflecting the proposed role of NOP-1 on the global activation of Rho.

*6) What are the phenotypic consequences of combined perturbations to *spd-1* and dynein? Indeed, this was previously reported by the Mishima group in Lee et al., (2015 PMID: 26088160) where *lin-5 RNAi* was shown to suppress the central-spindle defect of *spd-1 (oj5)* embryos. In that paper it was concluded that dynein-dependent cortical pulling forces contribute to the central spindle breakage observed upon loss of *spd-1* function. How can one separate dynein-dependent myosin transport from dynein-dependent spindle pulling forces, and other functions? This is crucial if a*

causative link between dynein-dependent myosin internalization and furrowing is to be established.

The influence of dynein inhibition on cytokinesis in the *spd-1(oj5)* or *spd-1(RNAi)* background is not so simple since it can act both positively and negatively. In the *spd-1*-defective embryos, although the central spindle is mechanically broken by the cortical pulling forces, centralspindlin is intact. As discussed above, in the embryos defective for SPD-1, cytokinesis is frequently completed as a combined effect of the enhanced polar relaxation by the asters localized more closely to the cortex. The re-routing of the liberated centralspindlin to the equatorial cortex by astral microtubules described above (Reviewer 3, point 3) might also contribute to the successful completion of cytokinesis. As we reported in Lee et al. (2015), depletion of LIN-5 alleviates the spindle defect of the *spd-1(oj5)* embryos and partially restores the midzone accumulation of centralspindlin; however, the suppression of the spindle defect is not perfect. In this case, the midzone centralspindlin signal is weaker than that in the wild-type embryos and starts to gradually disappear at ~90 s after anaphase onset (Lee et al. (2015) Figure 5). On the other hand, lack of the dynein-dependent removal of the cortical myosin II would strongly abolish the polar relaxation. A combination of the partial defect in the centralspindlin-dependent equatorial stimulation and the absence of the dynein-dependent polar relaxation would result in partial defects of cytokinesis. Indeed, 10 in 20 *spd-1(oj5);lin-5(RNAi)* embryos failed cytokinesis at 22°C due to furrow regression while only 1 in 17 *spd-1(oj5)* embryos did so (our unpublished observations). This is consistent with the previous report of the synthetic furrow regression phenotype in the *spd-1(oj5);gpr-1/2(RNAi)* embryos (Bringmann 2007; Verbrugghe 2007).

Figure 5 (Lee et al., Nat Commun 6, 7290 (2015))

Dynein is a multifunctional motor protein, which plays crucial roles in nearly every event in cell division. During anaphase and cytokinesis, dynein, LIN-5 and GPR-1/2 are responsible for spindle positioning and elongation, and removal of myosin II from the cell cortex. Especially when the cortical actomyosin network is broken (myosin II depletion, cytochalasin treatment), they also cause tube-like invaginations of the plasma membrane, which are morphologically distinct from the punctate signals of the myosin II particles we report in this work. Currently, it is difficult to clearly explain the relationships between these phenomena in molecular terms. As mentioned in the response to Reviewer 1, we observed the embryos expressing both the marker for

the plasma membrane (PH::GFP) and NMY-2::tagRFP-T and found that these are not always colocalized on cytoplasmic particles moving towards the spindle poles (current **Fig. 3 A** and **B**), implying that the relationships between dynein, myosin II and the plasma membrane are not so simple. This is in line with a recent report of the heterogeneous populations of dynein in the cell cortex and the microtubule tips (Schmidt et al. (2017), PMID: 28739679). To specifically dissect out the myosin transport from the other dynein functions, we need to reveal the molecular details of the linkage between myosin II and dynein and target a specific component on this linkage. This is beyond the scope of the current work.

7) In lin-5(RNAi) embryos (Fig 2c, Supp. Video 4), it is claimed that no cytoplasmic movement of NMY-2 was observed, yet clearing of myosin from the polar cortex can be seen, suggesting that myosin internalization is NOT the only mechanism responsible for its delocalization.

Thank you for pointing this out, we had noticed this but failed to mention it in our previous version. We mention this in the Discussion (page 20-21).

March 10, 2020

Re: JCB manuscript #201903080R

Dr. Masanori Mishima
University of Warwick
Warwick Medical School
Gibbet Hill Road
Coventry CV4 7AL
United Kingdom

Dear Masanori,

Thank you for submitting your revised manuscript entitled "Polar relaxation by dynein-mediated removal of cortical myosin II". Your manuscript has now been re-reviewed by two of the original three reviewers, whose full comments are appended below. Based on their comments, we will be willing to consider a revised version of the manuscript assuming you address all of the points raised by reviewer 1 (which do not require new experiments) and assuming you tone down your claims along the lines suggested by reviewer 2. Please note that I will expect to make a final decision without additional reviewer input upon resubmission.

Bill

please also attend to the following formatting changes:

- Please provide a short eTOC blurb
- Provide main and supplementary text as separate, editable .doc or .docx files
- Provide figures as separate, editable files according to the instructions for authors on JCB's website, paying particular attention to the guidelines for preparing images at sufficient resolution for screening and production
- Add scale bars to figures 2A, 5C, S1A, S2A,
- Provide tables as excel files
- Add a paragraph after the Materials and Methods section briefly summarizing the online supplementary materials (i.e. include supp fig legends as well as videos)
- Add conflict of interest statement to Acknowledgements section

Please submit the final revision within one month, along with a cover letter that includes a point by point response to the remaining reviewer comments.

Thank you for this interesting contribution to Journal of Cell Biology. You can contact me or the scientific editor listed below at the journal office with any questions, cellbio@rockefeller.edu or call (212) 327-8588.

Sincerely,

William Bement, Ph.D.
Monitoring Editor

Marie Anne O'Donnell, Ph.D.
Scientific Editor

Journal of Cell Biology

Reviewer #1 (Comments to the Authors (Required)):

This is a re-review of a manuscript by Chapa-y-Lazo and colleagues, "Polar relaxation by dynein-mediated removal of cortical myosin II". Therein they provide evidence that myosin II aggregates are transported off the cortex of the *C. elegans* zygote during cytokinesis. They offer evidence that myosin is removed in association with astral microtubule ends that reach the cortex in anaphase. They analyze the spatial correlation of myosin removal events, and use various mutants and depletions to support their case that a) myosin removal equates with local cortical tension release, and b) the spatial organization of the spindle and astral microtubules therefore translates into a spatial tension differential on the cortex, which they equate with the polar relaxation hypothesis of Wolpert and others.

While I might wish dearly never to read the phrases "polar relaxation" and "equatorial stimulation" again, I largely endorsed the original version of this manuscript. The authors have offered reasoned and credible responses to my previous criticisms. I feel this paper should be published in JCB with minor textual revisions described below.

Two general critiques I make of the text, requiring no new data whatsoever, are that 1) there should be more information about how many times and how consistently a particular observation was made, and 2) I feel the authors have collected a lot of good data that is consistent with their case, but not definitive (which is fine), and yet their rhetorical choices frequently overstate the extent to which their data prove one or another interpretation. I enumerate below the instances of both points that I felt were most important.

p. 5-6, Fig. 2A, B; it would be useful to have some idea of how consistent this effect of nocodazole is. Likewise for dynein and LIN-5 depletion. The events in question are rare enough that, for example, Fig. 1E is an overlay of no less than 22 embryos. Therefore, how many embryos were compared for nocodazole treatment versus control?

p. 6-7: I don't feel the interpretation of the cytokinetic delay caused by LIN-5 or GPR1/2 is exclusively earned. The authors say this "suggests a positive role" for the myosin removal. There could be any number of other explanations for this observation (e.g., pole separation, which previous authors have shown is promoted by GPR-1/2 dependent pulling forces, and which promotes cytokinetic furrow initiation).

p. 7-8: "These observations [that some membrane invaginations have myosin at the tips, and some don't] suggest that dynein and astral microtubules act on the plasma membrane and the myosin II particles through distinct mechanisms." It does? It seems to me it suggests that there's some membrane invagination along astral microtubules, possibly with an endosomal fate, and sometimes it hits a myosin patch and carries it in, whereafter the myosin dissociates or the membrane changes

or whatever. Would one not observe the same thing during endocytosis, using markers various of coat components?

Also, the authors should convey some sense of how many instances of one or the other kind were observed. I do not think it demands a statistical analysis; all I want to see is "we examined n PH/Myosin dual-labeled embryos and assessed x many particles, of which approximately a/x were Ph-positive and myosin-negative, and b/x were PH-negative and myosin-positive".

p. 9-10, Fig. 4B: A local minimum of microtubule density at the future furrow; fine. The same data show a global minimum at the anterior pole. As have others. I don't think the authors should sweep this under the rug; if they want to keep talking about poles and polar relaxation, they need to face the fact that the anterior pole (in these cells) is farther from the spindle poles and experiences lower astral microtubule penetration than any other part. This is the point of Rappaport's famous cylindrical sand dollar eggs, which he interpreted - erroneously - as a disproof of "polar" relaxation. *C. elegans* just happens to do this experiment to itself. Now, Rappaport erred by taking the semantics literally: mechanically, it matters not at all whether the hypothetical reduction of surface tension happens at the true poles of the cell or in its temperate (or even tropical!) latitudes. Importantly, in this comment I am not disputing the authors' data, just the words they use to describe it.

p. 10: "So far, we have shown that ... removal of myosin II from the cortex causes local reduction of the cortical tension/contractility". I do not feel the data presented in Figure 3 sufficiently demonstrate this conclusion. The data show two cases in which points on the cortex spread apart following departure of an intervening myosin particle. This is *consistent* with the hypothesis, but proof would require at least enough cases to show that most removals are associated with a subsequent spread, and ideally that spreading rate covaries with removal rate. It is a simple matter to change the wording so that the strength of the conclusion is not over-stated, and also to include a statement of the number of instances documented (and I apologize if I missed such a statement somewhere).

(Note: I want to be clear that in this and similar points, I am NOT demanding that the authors collect more data until they can make definitive, statistically-supported statements; rather, that their statements match the data they have, and that they be accompanied by information about how often they've made the observations in question. Are these two lucky catches? Do they have dozens of such cases? How uniform are they? Please just say in the text. I recognize the challenge of collecting these data, which require in this instance that a removal event be matched by two adjacent fiduciary marks, that the dominant tensile arc be aligned with the plane of focus, that the fiduciary marks persist, etc. It would therefore be unsurprising if not all recorded events conform to the hypothesis, even if it were fully and exclusively valid.)

p. 11: "Importantly, despite the disruption of the central spindle, an important source of a positive signal for the contractile ring assembly, the initiation of cleavage furrow formation was accelerated". Many previous authors have noted such effects; we have suggested (e.g., Baruni et al, *J. Cell Sci.* 121:306) that this is because the central spindle is not merely a source of the positive signal, but also sequesters it. I assume that the authors' next statements, about the effect of CYK-4 depletion, are meant to contradict this interpretation in favor of one which blames the acceleration on aster-dependent myosin depletion. OK, but I feel the argument is incomplete, as CYK-4 is in our interpretation a sequestering agent as well as an activator. Ect2 is clearly active at some level without CYK-4, at least via the NOP-1 dependent pathway if not otherwise too.

p. 14, bottom: again, I feel that the data is consistent with other explanations than the one the authors seek to support, and that their case would not be weakened at all by rephrasing to say, "this effect is consistent with our hypothesis that asters mediate cortical relaxation by removing myosin from the cortex, thus prompting flow towards the equator."

p. 18: "although our data do not exclude the possibility that dynein might also contribute to the equatorial stimulation by an unknown mechanism". Beyond the scope of this paper, but yes, indeed, observations like the ones herein make me wonder whether a similar mechanism might directly remove Ect2 or active Rho from the cortex.

I would also like to remark that the other two reviewer's critiques seem to set a very high bar for publication. In particular I don't think they can be expected to resolve the CYK-4 GAP debate, yet they also cannot be expected to pretend this factor doesn't exist, as if it has nothing to do with cytokinetic furrow specification until the Great GAP Gap has been bridged. I also don't think the authors should be expected to definitively dissect the many different functions of pleiotropic factors (e.g., dynein), but they also can't NOT use this experiment to try to support their case, even if by itself it is ambiguous. Many of my own minor comments reflect similar issues, all of which I think would be resolved if the authors adopted a slightly less conclusive tone for their interpretive statements.

Reviewer #3 (Comments to the Authors (Required)):

It is clear from the revised manuscript and rebuttal that substantive efforts have been made to address some of the concerns raised, which is appreciated. However, I still have major reservations as to the validity of the central claim that removal of cortical myosin II drives local polar relaxation and contributes to the initiation of cytokinesis. Dynein-dependent internalization/movement of myosin II particles does appear to be occurring but a causal link to polar relaxation and furrow initiation is still missing, and there are additional uncertainties, as discussed below.

I therefore still cannot support publication in its current form.

There are nice data worthy of publication, but the conclusions need to be greatly toned down, restricted to what the data actually show and the caveats in interpretation more openly discussed. It would be fine to finish with a discussion of the proposed model (which is certainly plausible), but it should not be presented as conclusively demonstrated, as is the case currently. Such a more cautiously worded manuscript, would be a valuable contribution to the literature, but may not meet JCB's high standards for a mechanistic advance.

Specific points:

- 1) There appear to be many cytoplasmic myosin-positive particles evident at metaphase (Video 1). Are they transported towards the spindle poles during anaphase? One cannot tell because the movie stops before anaphase onset, but how can one differentiate those pre-existing myosin particles from ones newly internalized during anaphase (of which there appear to be few clear examples of internalization)? Only newly internalized ones could contribute to the claimed polar relaxation, thus there would appear to be heterogeneous populations of myosin particles. In addition, from Fig. 3A-B (& videos 5 and 6), the discussion in the rebuttal letter, and the previous works of Tse et al. (2011) and Redemann et al. (2010), there is evidence of additional heterogeneity in terms of particle composition (myosin II-positive and negative, PH-positive, ANI-1-positive) that are internalized by dynein-dependent mechanisms. Even if dynein-dependent cortical relaxation is occurring, the other events have not been excluded, so it appears misleading to claim that it is

specifically due to myosin II internalization.

2) The claim of dynein-dependent cortical relaxation also does not appear adequately supported. Video 4 (*lin-5(RNAi)*) shows an absence of internalization of myosin II and pole-directed myosin particle motion as claimed. However, the anterior polar cortex appears to show dynamic patterns of myosin clearing consistent with local relaxation, similar to that shown in Fig. 3. I am not convinced that the internalization events shown in Fig. 3 (Videos 8 & 9), as was the case for Fig. 1d of the previous submission, directly lead to the observed separation of cortical puncta. It is an intriguing possibility, but it could also be correlation or chance. Do the separation of cortical puncta (which reportedly correlate with myosin internalization events) never occur when myosin internalization is blocked (e.g. dynein pathway inhibition)? If not, how confident can one be that any given internalization event is causal? Perhaps tight temporal correlation would increase confidence but such analysis is lacking.

3) Indeed, seeing the MTs probing the cortex (Videos 8 & 9), and given the role of MTs in the internalization events, also makes one wonder whether the lateral movement of cortical myosin patches (blue arrowheads) could not be influenced by interactions with the MTs, thereby calling into question the assumption that their separation reflects cortical relaxation.

4) As for the SPD-1 effects on the centralspindlin pathway, I think it is dangerous to assume that CYK4 is not involved because it is not detectable at the cortex (Reviewer Figure 1). Undetectable levels may be sufficient, as has been shown for ECT2 in human cells for example (Kotynkova et al, Cell Rep. 2016, PMID: 27926870). The new CYK-4 depletion data are nice but also not entirely conclusive because any residual protein (even undetectable) may be enough to initiate furrowing despite clear disruption of the central spindle. SPD-1 and CYK-4 co-depletion might provide a better test of whether CYK-4 is still contributing.

5) The NOP-1 data are a nice addition.

Our response to the Reviewers' comments

Original Reviewers' comments

Our response

Reviewer #1 (Comments to the Authors (Required)):

*This is a re-review of a manuscript by Chapa-y-Lazo and colleagues, "Polar relaxation by dynein-mediated removal of cortical myosin II". Therein they provide evidence that myosin II aggregates are transported off the cortex of the *C. elegans* zygote during cytokinesis. They offer evidence that myosin is removed in association with astral microtubule ends that reach the cortex in anaphase. They analyze the spatial correlation of myosin removal events, and use various mutants and depletions to support their case that a) myosin removal equates with local cortical tension release, and b) the spatial organization of the spindle and astral microtubules therefore translates into a spatial tension differential on the cortex, which they equate with the polar relaxation hypothesis of Wolpert and others.*

While I might wish dearly never to read the phrases "polar relaxation" and "equatorial stimulation" again, I largely endorsed the original version of this manuscript. The authors have offered reasoned and credible responses to my previous criticisms. I feel this paper should be published in JCB with minor textual revisions described below.

Two general critiques I make of the text, requiring no new data whatsoever, are that 1) there should more information about how many times and how consistently a particular observation was made, and 2) I feel the authors have collected a lot of good data that is consistent with their case, but not definitive (which is fine), and yet their rhetorical choices frequently overstate the extent to which their data prove one or another interpretation. I enumerate below the instances of both points that I felt were most important.

We appreciate this reviewer's supportive comments and constructive criticisms. In this revision, we included numbers that were missing in the previous versions and modified the text according to their suggestions as detailed below.

R1_1)

p. 5-6, Fig. 2A, B; it would be useful to have some idea of how consistent this effect of nocodazole is. Likewise for dynein and LIN-5 depletion. The events in question are rare enough that, for example, Fig. 1E is an overlay of no less than 22 embryos. Therefore, how many embryos were compared for nocodazole treatment versus control?

After testing different concentrations of nocodazole, we treated 5 embryos with 15 μ M nocodazole and observed a nearly immediate (within \sim 30 s) stop of movement of the myosin II particles in all cases while no suspension of movement was observed in 4 embryos treated with DMSO.

We observed 7 *dhc-1(RNAi)* embryos along with 3 control RNAi embryos and 33 *lin-*

5(RNAi) embryos with 33 controls. All of the *dhc-1(RNAi)* embryos showed neither internalization nor centrosome-directed movement of myosin II particles. No unidirectional movement of the cytoplasmic myosin II particles was observed in any of the 33 *lin-5(RNAi)* embryos although a small number of immobile cytoplasmic particles were observed in 11 of them. This was in stark contrast with the control embryos, in which we always detected some particles moving unidirectionally towards the centrosomes.

These numbers are indicated in pages 8 and 9 (marked yellow) with a reference to the quantitative analysis in Fig.7 C.

R1_2)

p. 6-7: I don't feel the interpretation of the cytokinetic delay caused by LIN-5 or GPR1/2 is exclusively earned. The authors say this "suggests a positive role" for the myosin removal. There could be any number of other explanations for this observation (e.g., pole separation, which previous authors have shown is promoted by GPR-1/2 dependent pulling forces, and which promotes cytokinetic furrow initiation).

We modified the text "These data are consistent with a positive role..." (page 9).

R1_3)

p. 7-8: "These observations [that some membrane invaginations have myosin at the tips, and some don't] suggest that dynein and astral microtubules act on the plasma membrane and the myosin II particles through distinct mechanisms." It does? It seems to me it suggests that there's some membrane invagination along astral microtubules, possibly with an endosomal fate, and sometimes it hits a myosin patch and carries it in, whereafter the myosin dissociates or the membrane changes or whatever. Would one not observe the same thing during endocytosis, using markers various of coat components?

Although we once considered the scenario suggested by this reviewer, we thought that the moving myosin II particles without the membrane signal indicated a distinct mechanism that does not require a membrane vesicle as a linker between dynein and myosin II. However, now we have realized that the absence of the signal of PH-domain does not necessarily mean the absence of a membrane vesicle since, for example, PIP2 might be lost after endocytosis. We modified the sentence to "According to these observations, it is likely that ..." (page 10).

Also, the authors should convey some sense of how many instances of one or the other kind were observed. I do not think it demands a statistical analysis; all I want to see is "we examined n PH/Myosin dual-labeled embryos and assessed x many particles, of which approximately a/x were Ph-positive and myosin-negative, and b/x were PH-negative and myosin-positive".

As suggested, we have now added the sentence: "We examined 8 PH/Myosin dual-labeled embryos and observed 20 myosin II particles negative for the PH domain signal, 31 membrane signals positive for the PH domain but negative for myosin II,

and 8 double positive particles.” in page 10.

R1_4)

p. 9-10, Fig. 4B: A local minimum of microtubule density at the future furrow; fine. The same data show a global minimum at the anterior pole. As have others. I don't think the authors should sweep this under the rug; if they want to keep talking about poles and polar relaxation, they need to face the fact that the anterior pole (in these cells) is farther from the spindle poles and experiences lower astral microtubule penetration than any other part. This is the point of Rappaport's famous cylindrical sand dollar eggs, which he interpreted - erroneously - as a disproof of "polar" relaxation. C. elegans just happens to do this experiment to itself. Now, Rappaport erred by taking the semantics literally: mechanically, it matters not at all whether the hypothetical reduction of surface tension happens at the true poles of the cell or in its temperate (or even tropical!) latitudes. Importantly, in this comment I am not disputing the authors' data, just the words they use to describe it.

We now clarified the presence of a global minimum at the anterior tip of the embryo in page 12 by saying: “Reflecting the posterior shift of the spindle after anaphase onset, the density of the microtubules first started to increase at the posterior cortex, creating a global minimum at the anterior tip of the embryo”.

R1_5)

*p. 10: "So far, we have shown that ... removal of myosin II from the cortex causes local reduction of the cortical tension/contractility". I do not feel the data presented in Figure 3 sufficiently demonstrate this conclusion. The data show two cases in which points on the cortex spread apart following departure of an intervening myosin particle. This is *consistent* with the hypothesis, but proof would require at least enough cases to show that most removals are associated with a subsequent spread, and ideally that spreading rate covaries with removal rate. It is a simple matter to change the wording so that the strength of the conclusion is not over-stated, and also to include a statement of the number of instances documented (and I apologize if I missed such a statement somewhere).*

(Note: I want to be clear that in this and similar points, I am NOT demanding that the authors collect more data until they can make definitive, statistically-supported statements; rather, that their statements match the data they have, and that they be accompanied by information about how often they've made the observations in question. Are these two lucky catches? Do they have dozens of such cases? How uniform are they? Please just say in the text. I recognize the challenge of collecting these data, which require in this instance that a removal event be matched by two adjacent fiduciary marks, that the dominant tensile arc be aligned with the plane of focus, that the fiduciary marks persist, etc. It would therefore be unsurprising if not all recorded events conform to the hypothesis, even if it were fully and exclusively valid.)

We have revised the sentence to read “So far, we have shown that astral microtubules and dynein remove myosin II preferentially from the polar/non-equatorial cortexes and that removal of myosin II from the cortex is frequently associated with local reduction of the cortical tension/contractility” (page 13).

We have also added the sentence “Although it was difficult to systematically and statistically assess the frequency of this phenomenon, in 20 embryos, we could detect 39 cases of internalization, 30 of which were followed by local cortical relaxation.” where we describe the association between the removal and relaxation (page 11).

R1_6)

p. 11: "Importantly, despite the disruption of the central spindle, an important source of a positive signal for the contractile ring assembly, the initiation of cleavage furrow formation was accelerated". Many previous authors have noted such effects; we have suggested (e.g., Baruni et al, J. Cell Sci. 121:306) that this is because the central spindle is not merely a source of the positive signal, but also sequesters it. I assume that the authors' next statements, about the effect of CYK-4 depletion, are meant to contradict this interpretation in favor of one which blames the acceleration on aster-dependent myosin depletion. OK, but I feel the argument is incomplete, as CYK-4 is in our interpretation a sequestering agent as well as an activator. Ect2 is clearly active at some level without CYK-4, at least via the NOP-1 dependent pathway if not otherwise too.

While it has been repeatedly reported that neither disruption of the central spindle nor depletion of the centralspindlin components prevent initial furrow formation, to our knowledge, acceleration of the furrow initiation has not been reported until very recently (Adriaans et al., 2019). In Discussion, we mention the possibility that the central spindle sequesters an unknown signal, “however, we cannot exclude the possibility that a positive signal is released from the central spindle upon its rupture (Baruni et al., 2008; Adriaans., 2019) and delivered to the cortex via astral microtubules by an unknown motor protein (Atilgan et al., 2012) earlier than normal.” (page 23) .

R1_7)

p. 14, bottom: again, I feel that the data is consistent with other explanations than the one the authors seek to support, and that their case would not be weakened at all by rephrasing to say, "this effect is consistent with our hypothesis that asters mediate cortical relaxation by removing myosin from the cortex, thus prompting flow towards the equator."

The modification was made, accordingly. Now the sentence reads “These observations are consistent with our hypothesis that asters mediate cortical relaxation by removing myosin II from the non-equatorial cortex, thus prompting flow towards the equator” (page 18).

R1_8)

p. 18: "although our data do not exclude the possibility that dynein might also contribute to the equatorial stimulation by an unknown mechanism". Beyond the scope of this paper, but yes, indeed, observations like the ones herein make me wonder whether a similar mechanism might directly remove Ect2 or active Rho from the cortex.

I would also like to remark that the other two reviewer's critiques seem to set a very

high bar for publication. In particular I don't think they can be expected to resolve the CYK-4 GAP debate, yet they also cannot be expected to pretend this factor doesn't exist, as if it has nothing to do with cytokinetic furrow specification until the Great GAP Gap has been bridged. I also don't think the authors should be expected to definitively dissect the many different functions of pleiotropic factors (e.g., dynein), but they also can't NOT use this experiment to try to support their case, even if by itself it is ambiguous. Many of my own minor comments reflect similar issues, all of which I think would be resolved if the authors adopted a slightly less conclusive tone for their interpretive statements.

We appreciate this reviewer's thoughtful comments. We believe that, in this revised version, our interpretive statements have been better adjusted to our experimental results.

Reviewer #3 (Comments to the Authors (Required)):

It is clear from the revised manuscript and rebuttal that substantive efforts have been made to address some of the concerns raised, which is appreciated. However, I still have major reservations as to the validity of the central claim that removal of cortical myosin II drives local polar relaxation and contributes to the initiation of cytokinesis. Dynein-dependent internalization/movement of myosin II particles does appear to be occurring but a causal link to polar relaxation and furrow initiation is still missing, and there are additional uncertainties, as discussed below.

I therefore still cannot support publication in its current form.

There are nice data worthy of publication, but the conclusions need to be greatly toned down, restricted to what the data actually show and the caveats in interpretation more openly discussed. It would be fine to finish with a discussion of the proposed model (which is certainly plausible), but it should not be presented as conclusively demonstrated, as is the case currently. Such a more cautiously worded manuscript, would be a valuable contribution to the literature, but may not meet JCB's high standards for a mechanistic advance.

We have modified the text as detailed above in our response to Reviewer #1. We believe that it reflects our experimental results more precisely.

Specific points:

R3_1)

1) There appear to be many cytoplasmic myosin-positive particles evident at metaphase (Video 1). Are they transported towards the spindle poles during anaphase? One cannot tell because the movie stops before anaphase onset, but how can one differentiate those pre-existing myosin particles from ones newly internalized during anaphase (of which there appear to be few clear examples of internalization)? Only newly internalized ones could contribute to the claimed polar relaxation, thus there would appear to be heterogeneous populations of myosin particles. In addition, from Fig. 3A-B (& videos 5 and 6), the discussion in the rebuttal letter, and the previous works of Tse et al. (2011) and Redemann et al. (2010), there is evidence of additional heterogeneity in terms of particle composition (myosin II-positive and negative, PH-positive, ANI-1-positive) that are internalized by dynein-

dependent mechanisms. Even if dynein-dependent cortical relaxation is occurring, the other events have not been excluded, so it appears misleading to claim that it is specifically due to myosin II internalization.

We regret that we failed to describe the fate of the cytoplasmic particles that are observed before anaphase onset but don't show unidirectional motility towards the spindle poles/centrosomes in the previous versions. As shown in the new Video 2, such particles exist before mitotic entry/nuclear envelop breakdown, gradually disappear during prometaphase and metaphase, and are cleared up from the cytoplasm at the anaphase onset. The majority of the anaphase particles that exhibit unidirectional movement towards the spindle poles seem to have departed from the cortex. In this revision, we described the fate of the pre-anaphase particles referring to the new Video 2, "NMY-2::GFP was also observed as cytoplasmic particulate signals, which showed diffusive random motion and gradually disappeared during early mitosis and were nearly undetectable at the metaphase to anaphase transition (Fig. 1A metaphase, Videos 1 and 2)" (page 7).

To clarify that we are not excluding the possible roles of other factors, we modified the sentence summarizing the paragraph that mentions the heterogeneity of the particles/vesicles to read "Although it is possible that internalization of other factors might play a role in the regulation of cytokinesis, we focused on the removal of the myosin II particles from the cell cortex in this work." (page 10). We also modified a sentence discussing the relations of the myosin transport and the other known phenomena driven by dynein to read "(various processes known to be driven by dynein) are involved in dynein-driven transport of myosin II and regulation of other cortical activities (and, vice versa) will be important future questions" (page 24).

R3_2)

2) The claim of dynein-dependent cortical relaxation also does not appear adequately supported. Video 4 (lin-5(RNAi)) shows an absence of internalization of myosin II and pole-directed myosin particle motion as claimed. However, the anterior polar cortex appears to show dynamic patterns of myosin clearing consistent with local relaxation, similar to that shown in Fig. 3.

Please note that Video 4 (now Video 5) starts at the timing of furrow ingression, which is significantly (~50 s or more) delayed in *lin-5(RNAi)* embryos (Fig. 2 C). It is not appropriate to compare this with the normal embryos shown in Fig. 3. In the embryos depleted of GPR-1/2, the partner of LIN-5, posterior-directed cortical flow is observed all over the cortex (especially in the anterior cortex) shortly after anaphase onset (Fig. 7A iii). We speculate that the myosin clearing from the anterior cortex is a consequence of this flow, which is driven by the myosin that has over-accumulated at the posterior cortex (Fig. 7B).

I am not convinced that the internalization events shown in Fig. 3 (Videos 8 &9), as was the case for Fig. 1d of the previous submission, directly lead to the observed separation of cortical puncta. It is an intriguing possibility, but it could also be correlation or chance. Do the separation of cortical puncta (which reportedly correlate with myosin internalization events) never occur when myosin internalization

is blocked (e.g. dynein pathway inhibition)? If not, how confident can one be that any given internalization event is causal? Perhaps tight temporal correlation would increase confidence but such analysis is lacking.

We agree that our observations are not strong enough to unequivocally conclude the causal relation. Thus, we modified the sentence summarizing them to “So far, we have shown that astral microtubules and dynein remove myosin II preferentially from the polar/non-equatorial cortexes and that removal of myosin II from the cortex is frequently associated with local reduction of the cortical tension/contractility” (page 13).

As Reviewer #1 pointed out, it is not trivial to assess the one-to-one relation between the internalization and the local relaxation in a statistically rigorous manner since it requires that both the responsible microtubule and the main axis of the relaxation are in the focal plane. We can, however, examine the divergent cortical flow as a proxy of the relaxation, although it is reflecting a collective effect of multiple events of the internalization and the local relaxation. A posterior-to-anterior flow near the posterior tip indicates a divergent flow from the posterior tip of the cell. In wild type embryos, this starts at 20 s after anaphase onset (Fig. 7A i). By contrast, in *gpr-1/2(RNAi)* embryos, this was not observed until 100 s and it was severely weakened (Fig. 7A iii). Importantly, this late and weak posterior-to-anterior flow (a divergent flow at the posterior tip) is almost completely suppressed by the loss of the equatorial stimulation (Fig. 7A iv, *cyk-4(GAP) gpr-1/2(RNAi)*), indicating that the late divergent flow away from the posterior tip is one triggered by the equatorial contraction.

R3_3)

3) Indeed, seeing the MTs probing the cortex (Videos 8 & 9), and given the role of MTs in the internalization events, also makes one wonder whether the lateral movement of cortical myosin patches (blue arrowheads) could not be influenced by interactions with the MTs, thereby calling into question the assumption that their separation reflects cortical relaxation.

The influence of microtubules on the lateral cortical motion of myosin patches is an interesting idea. However, in Video 9, the patch marked with a blue arrowhead on the top side is moving upwards while the bulk of microtubules nearby are moving downwards. Thus, the suggested mechanism does not easily explain that observation.

R3_4)

4) As for the SPD-1 effects on the centralspindlin pathway, I think it is dangerous to assume that CYK4 is not involved because it is not detectable at the cortex (Reviewer Figure 1). Undetectable levels may be sufficient, as has been shown for ECT2 in human cells for example (Kotynkova et al, Cell Rep. 2016, PMID: 27926870). The new CYK-4 depletion data are nice but also not entirely conclusive because any residual protein (even undetectable) may be enough to initiate furrowing despite clear disruption of the central spindle. SPD-1 and CYK-4 co-depletion might provide a better test of whether CYK-4 is still contributing.

Kotynkova et al. demonstrated that an ECT2 mutant defective for the interaction with CYK4/RACGAP1 can induce furrowing although it fails to localize to the spindle midzone. In this case, however, the mutant protein is still recruited to the cell cortex with slight enrichment to the furrow. We don't think this is an appropriate example of the sufficient activity of an undetectable level of a protein. Nevertheless, it is possible that an undetectable level of CYK4 released from the central spindle contributes to the accelerated furrow initiation. We discussed the possibility of release of a positive signal from the ruptured central spindle in Discussion (page 23).

Co-depletion of SPD-1 and CYK-4 would be an interesting experiment, but whatever the result, we wouldn't be able to reach an absolute conclusion if the same argument of the sufficient activity of an undetectable level of proteins is applied. Unfortunately, due to the national lockdown because of COVID-19, we cannot perform such an experiment in the near future.

5) The NOP-1 data are a nice addition.